# Causal links of α-thalassemia indices and cardiometabolic traits and diabetes: MR study

Lung-An Hsu[1], Semon Wu[2], Ming-Sheng Teng[3], Yu-Lin Ko[3,4,5]

Our study aimed to investigate if genetic variants around 16p13.3's *HBA1* locus, associated with erythrocyte indices and HbA1c levels, predict α-thalassemia-related erythrocyte indices, cardiometabolic traits, and diabetes risk in Taiwanese individuals. We analyzed Taiwan Biobank data, including whole-genome sequencing from 1,493 participants and genotyping arrays from 129,542 individuals. First, we performed regional association analysis using whole-genome sequencing data to identify genetic variants significantly associated with erythrocyte indices, confirming their linkage disequilibrium with the α⁰ thalassemia $-^{SEA}$ deletion mutation, a common cause of α-thalassemia in Southeast Asian populations. Deletion mutation sequencing further validated these variants' association with α-thalassemia. Subsequently, we analyzed genotyping array data, revealing associations between specific genetic variants and cardiometabolic traits, including lipid profiles, HbA1c levels, bilirubin levels, and diabetes risk. Using Mendelian randomization, we established causal relationships between α-thalassemia-related erythrocyte indices and cardiometabolic traits, elucidating their role in diabetes susceptibility. Our findings highlight genetic variants around the α-globin genes as surrogate markers for common α-thalassemia mutations in Taiwan, emphasizing the causal links between α-thalassemia-related erythrocyte indices, cardiometabolic traits, and heightened diabetes risk.

## Introduction

Microcytic anemia is the most common type of anemia in both childhood and adulthood (Richardson, 2007; DeLoughery, 2014; Urrechaga et al, 2015; Zanetti et al, 2021). Microcytic anemia is characterized by the presence of small-sized erythrocytes because of the decreased availability of hemoglobin components, such as globins, iron, and heme; this, in turn, causes a reduction in the hemoglobin content in RBC precursors, subsequently resulting in delayed erythroid differentiation (DeLoughery, 2014; Cappellini et al, 2020). Microcytic anemia may be caused by genetic or non-genetic factors, and three subgroups of inherited microcytic anemia have been classified based on the causative defects: hemoglobinopathies (thalassemia) caused by defects in globin chain synthesis, defects in heme synthesis, and defects in iron availability or acquisition by erythroid precursors (DeLoughery, 2014; Cappellini et al, 2020). Although most of the genetic causes of microcytic anemia are rare in terms of occurrence and each mutation contributes to a small proportion of patients with microcytic anemia, some mutations may be dominant in distinct ethnic populations, such as α⁰ thalassemia deletion $-^{SEA}$, $-^{Fil}$, and $-^{THAI}$ in the Southeast Asian region (Chen et al, 2002; Iolascon et al, 2009; Harteveld & Higgs, 2010; Chao et al, 2014; Lee et al, 2015; Wang et al, 2017; Farashi & Harteveld, 2018; Mettananda & Higgs, 2018; Cappellini et al, 2020).

Thalassemia is among the most common human monogenic disorders (Weatherall & Clegg, 1996). α-thalassemia, which is caused by defective α-globin gene mutations on chromosome 16p13.3, is a hemoglobinopathy characterized by deficits in α-globin chain synthesis and microcytic hypochromic erythrocytes (Harteveld & Higgs, 2010; Higgs et al, 2012; Piel & Weatherall, 2014; Mettananda & Higgs, 2018; Taher et al, 2018). Recently, Mendelian randomization (MR) studies have reported that genetically predicted erythrocyte indices are causally related to cognitive function deficits, Alzheimer's disease, and venous thromboembolism (Winchester et al, 2018; Luo et al, 2020). Multiple erythrocyte indices were identified to be associated with HbA1c levels and diabetes mellitus (DM) development (Wang et al, 2021). Genome-wide association studies (GWASs) have revealed that chromosome 16p13.3 as a gene locus is associated with erythrocyte traits and hemoglobin A1c levels (Kichaev et al, 2019; Vuckovic et al, 2020; Sinnott-Armstrong et al, 2021). However, causal relationships between α-thalassemia-related erythrocyte indices and cardiometabolic traits remain to be completely elucidated. The Taiwan Biobank (TWB) is a large-scale population-based cohort study including individuals aged between 30 and 70 yr who have no history of cancer (Fan et al, 2008; Chen et al, 2016). More than 120,000 participants in the TWB study underwent whole-genome genotyping array analysis, and a subgroup of them also underwent whole-

[1]The First Cardiovascular Division, Department of Internal Medicine, Chang Gung Memorial Hospital and Chang Gung University College of Medicine, Taoyuan, Taiwan [2]Department of Life Science, Chinese Culture University, Taipei, Taiwan [3]Department of Research, Taipei Tzu Chi Hospital, Buddhist Tzu Chi Medical Foundation, New Taipei City, Taiwan [4]The Division of Cardiology, Department of Internal Medicine, Taipei Tzu Chi Hospital, Buddhist Tzu Chi Medical Foundation, New Taipei City, Taiwan [5]School of Medicine, Tzu Chi University, Hualien, Taiwan

Correspondence: yulinkotw@yahoo.com.tw

genome sequencing (WGS) analysis. Our study was primarily designed to uncover and elucidate the causal relationships between α-thalassemia-related erythrocyte indices, cardiometabolic traits, and DM. We initiated by conducting a regional association analysis within the 16p13.3 region and performed deletion mutation sequencing on WGS data to pinpoint genetic variants serving as proxies for α-thalassemia. Subsequently, our investigation expanded to encompass genotyping array data from an extensive cohort of 129,542 participants. This enabled us to comprehensively explore associations between specific genetic variants and a spectrum of cardiometabolic traits, along with assessing the risk of DM. In the final phase, we harnessed MR to estimate the causal effect of α-thalassemia-related erythrocyte indices on various cardiometabolic traits and DM, thereby enhancing our understanding of these intricate interactions.

## Results

### Baseline characteristics of TWB participants

Fig 1 depicts the flowchart of participant enrollment with whole-genome genotyping array and WGS data. Table 1 summarizes the demographic data, clinical and biochemical data, and lipid profiles of TWB participants with WGS and GWAS array data. The average ages of participants selected for WGS and GWAS array analysis were 50 and 51 yr of age, respectively. The numbers of men and women were equal for participants with WGS data, whereas the number of women was higher for participants with array data.

### Regional association study for RBC parameters

We analyzed the association between genetic variants at positions between 0.06 and 0.68 Mb on chromosome 16p13.3 and the RBC parameters in TWB participants with WGS data. Our findings revealed that three single-nucleotide variations (SNVs), namely *NPRL3* rs191086839, *LUC7L* rs372755452, and *PGAP6* rs375498857, were specific to Asians (with minor allele frequencies ranging from 0.0177 to 0.0218 among TWB and East Asian populations versus all <0.0001 among other ethnic populations) (Table S1) and significantly associated with four RBC traits, namely the erythrocyte count, Hb level, mean corpuscular volume (MCV), and mean corpuscular hemoglobin (MCH), with the lowest *P*-values for each variant being $8.68 \times 10^{-93}$, $2.30 \times 10^{-17}$, $1.98 \times 10^{-101}$, and $1.76 \times 10^{-122}$, respectively (Fig 2). MCV and MCH were thus selected for in-depth examination with cardiometabolic traits based on their most significant association with 16p13.3 and their impact on erythrocyte count and Hb level. The LD map revealed strong LD among these three variants (Fig 3) in our study population.

### Association of the three variants on chromosome 16p13.3 with RBC parameters

In total, the data of 1,493 TWB volunteers were included in the study on the association of genotypes and phenotypes with RBC parameters (Table 2 and Fig 4). By performing general linear

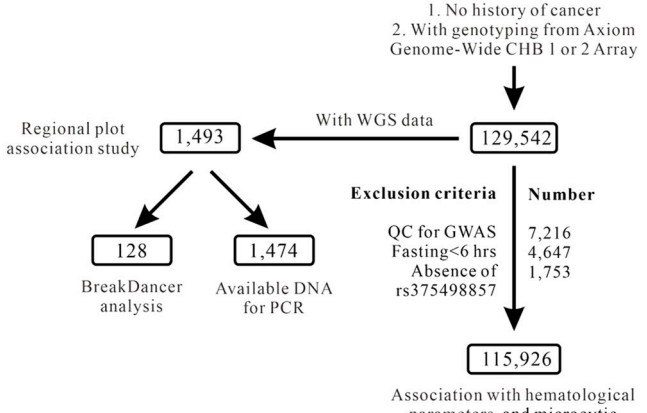

**Figure 1.  Flowchart of inclusion and exclusion criteria used to screen Taiwan Biobank project participants.**

Abbreviations: WGS, whole-genome sequence; GWAS, genome-wide association study; QC, quality control.

regression by using an additive model, we determined that after adjustment for age, sex, body mass index (BMI), and smoking status, participants harboring the minor allele of the three variants (i.e., the C allele for *NPRL3* rs191086839, a single-nucleotide deletion for *LUC7L* rs372755452, and the A allele for *PGAP6* rs375498857) exhibited a genome-wide significant association ($P < 5 \times 10^{-8}$) with a higher erythrocyte count and lower Hb, MCV, MCH, and mean corpuscular hemoglobin concentration values and a subthreshold significant association ($P$ between $5 \times 10^{-8}$ and $1 \times 10^{-4}$) with lower HCT values (Table 2). Furthermore, participants harboring the minor allele of each variant had significantly higher risks of the microcytic hypochromic trait, anemia, and microcytic anemia (Table 2 and Fig 4A–I). Stepwise linear regression analysis for MCV and MCH revealed that the association with these erythrocyte traits markedly diminished for the *PGAP6* rs375498857 genotype, but not for the other two variants (Table 3). These results suggest that the association between *PGAP6* rs375498857 and the erythrocyte traits is because of the strong LD with *NPRL3* rs191086839 and *LUC7L* rs372755452. Through a logistic regression analysis of all the three variants, we determined a significant association of *NPRL3* rs191086839 and *LUC7L* rs372755452 with the risk of the microcytic hypochromic trait (odds ratio [OR] = 40.87 and 20.2; 95% confidence interval [CI] = 7.26–229.99 and 2.13–191.3, respectively; Table S2) and of *NPRL3* rs191086839 with the risk of microcytic anemia (ORs = 132.74; 95% CIs = 11.80–1,493.49). These results revealed that the *NPRL3* rs191086839 variant was independently associated with all the erythrocyte traits analyzed.

### Searching for deletion mutations in the α-globin gene region in participants with WGS data: BreakDancer and PCR analysis with direct DNA sequencing

α-thalassemia is most likely caused by large deletion mutations in the α-globin gene (Fig 5A and B) (Richardson, 2007; DeLoughery, 2014; Urrechaga et al, 2015; Zanetti et al, 2021). Thus, we first used

**Table 1.** Baseline characteristics of the Taiwan Biobank participants with whole-genome sequencing and genome-wide association study array data.

| Clinical and laboratory parameters | Participants with WGS | | Participants with GWAS array data | |
|---|---|---|---|---|
| | Median (IQR) | Number | Median (IQR) | Number |
| Anthropology | | | | |
| Age, years | 50.0 (40.0–59.0) | 1,493 | 51.0 (41.0–59.0) | 115,926 |
| Sex, male/female | 50.0%/50.0% | 746/747 | 36.0%/64.0% | 41,752/74,174 |
| Waist circumference, cm | 84.0 (77.0–90.0) | 1,492 | 83.0 (76.0–90.0) | 115,863 |
| Waist–hip ratio | 0.87 (0.82–0.91) | 1,492 | 0.87 (0.82–0.91) | 115,859 |
| Body mass index, kg/m$^2$ | 23.9 (21.8–26.4) | 1,492 | 23.8 (21.6–26.3) | 115,847 |
| Blood pressure | | | | |
| Systolic BP*, mmHg | 112.5 (104.0–124.0) | 1,306 | 115.5 (105.5–127.0) | 101,916 |
| Diastolic BP*, mmHg | 71.0 (64.0–79.0) | 1,306 | 71.0 (65.0–79.0) | 101,916 |
| Mean BP*, mmHg | 85.0 (78.0–93.3) | 1,306 | 86.2 (78.8–94.6) | 101,916 |
| Lipid profiles | | | | |
| Total cholesterol#, mg/dl | 191.0 (170.0–215.0) | 1,394 | 193.0 (171.0–217.0) | 107,233 |
| HDL-cholesterol#, mg/dl | 52.0 (44.0–63.0) | 1,394 | 53.0 (45.0–63.0) | 107,233 |
| LDL-cholesterol#, mg/dl | 119.0 (101.0–141.0) | 1,394 | 119.0 (99.0–141.0) | 107,233 |
| Triglyceride#, mg/dl | 90.0 (64.0–131.0) | 1,394 | 91.0 (64.0–133.0) | 107,233 |
| Glucose metabolism | | | | |
| Fasting plasma glucose**, mg/dl | 92.0 (87.0–98.0) | 1,426 | 92.0 (87.0–97.0) | 109,997 |
| HbA1C**, % | 5.6 (5.4–5.8) | 1,426 | 5.6 (5.4–5.8) | 109,996 |
| Uric acid | | | | |
| Uric acid***, mg/dl | 5.4 (4.5–6.4) | 1,420 | 5.2 (4.4–6.2) | 111,451 |
| Renal function | | | | |
| Creatinine, mg/dl | 0.72 (0.59–0.87) | 1,493 | 0.68 (0.58–0.84) | 115,922 |
| eGFR, mL/min/1.73 m$^2$ | 100.7 (88.0–117.1) | 1,493 | 100.1 (86.9–115.6) | 115,913 |
| Albuminuria, mg/liter | 9.0 (5.5–15.8) | 1,491 | 8.7 (5.3–15.2) | 115,729 |
| Liver function related | | | | |
| AST, U/liter | 23.0 (19.0–27.0) | 1,493 | 23.0 (20.0–27.0) | 115,924 |
| ALT, U/liter | 19.0 (14.0–27.0) | 1,493 | 19.0 (14.0–27.0) | 115,764 |
| gGT, U/liter | 18.0 (12.0–27.0) | 1,493 | 17.0 (12.0–26.0) | 115,910 |
| Serum albumin, g/dl | 4.6 (4.4–4.7) | 1,493 | 4.5 (4.4–4.7) | 115,924 |
| Total bilirubin, mg/dl | 0.6 (0.5–0.8) | 1,493 | 0.6 (0.5–0.8) | 115,924 |
| Hematological parameters | | | | |
| Leukocyte count, 10$^3$/µl | 5.8 (5.0–6.9) | 1,493 | 5.6 (4.7–6.7) | 115,915 |
| Hematocrit, % | 43.7 (40.5–46.6) | 1,493 | 41.5 (38.9–44.4) | 115,915 |
| Platelet count, 10$^3$/µl | 232.0 (199.5–268.0) | 1,493 | 238.0 (202.0–277.0) | 115,913 |
| RBC count, 10$^6$/µl | 4.8 (4.4–5.1) | 1,493 | 4.7 (4.4–5.1) | 115,915 |
| Hemoglobin, g/dl | 14.1 (13.0–15.1) | 1,493 | 13.7 (12.8–14.8) | 115,915 |
| MCH, pg/RBC | 29.8 (28.7–30.8) | 1,493 | 29.7 (28.6–30.7) | 115,915 |
| MCHC, g/dl | 32.4 (31.1–33.4) | 1,493 | 33.3 (32.3–34.1) | 115,915 |
| MCV, fl | 91.8 (87.7–95.8) | 1,493 | 89.0 (85.6–92.2) | 115,915 |
| Atherosclerotic risk factors | | | | |
| Diabetes mellitus, % | 9.11% | 136 | 9.46% | 10,965 |
| Hypertension, % | 19.56% | 292 | 22.28% | 25,832 |

**Table 1. Continued**

| Clinical and laboratory parameters | Participants with WGS | | Participants with GWAS array data | |
|---|---|---|---|---|
| | Median (IQR) | Number | Median (IQR) | Number |
| Current smoking, % | 10.05% | 150 | 19.67% | 22,801 |
| Gout, % | 4.89% | 73 | 3.86% | 4,472 |
| Microalbuminuria, % | 11.74% | 175 | 11.20% | 12,969 |
| Metabolic syndrome, % | 19.30% | 288 | 25.53% | 29,592 |

Abbreviations: WGS, whole-genomic sequence; IQR, Inter-quartile range; BP, blood pressure; HbA1c, hemoglobin A1C; eGFR, estimated glomerular filtration rate; AST, aspartate aminotransferase; ALT, alanine aminotransferase; γGT, γ-glutamyl transferase; MCH, mean corpuscular hemoglobin; MCHC, mean corpuscular hemoglobin concentration; MCV, mean corpuscular volume. Participants were analyzed after the exclusion of those with a history of *hypertension, **diabetes mellitus, ***gout, and #hyperlipidemia.

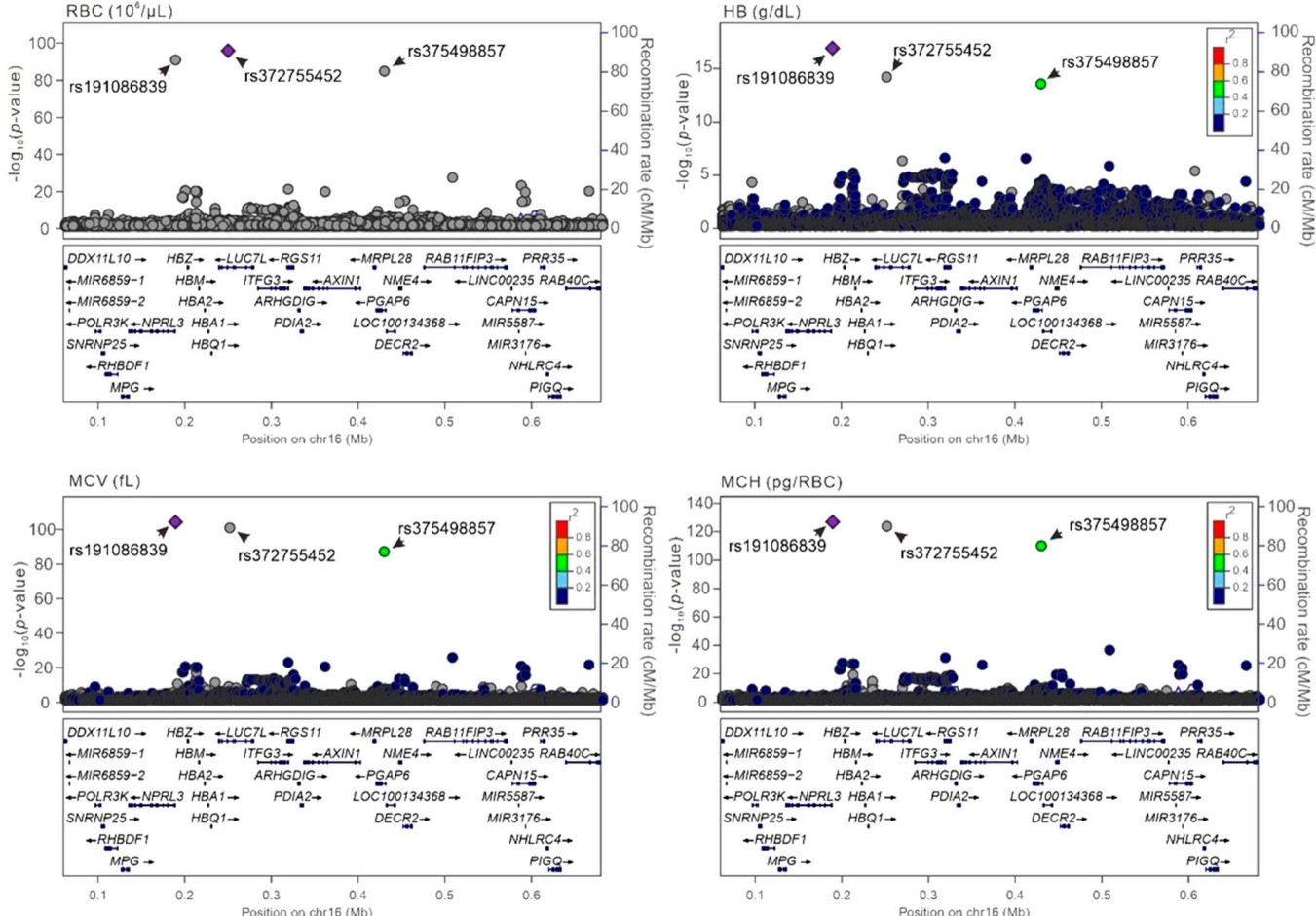

**Figure 2. Regional association study of RBC count, hemoglobin (Hb) level, mean corpuscular volume, and mean corpuscular hemoglobin around the chromosome16p13.3 regions in 1,493 Taiwan Biobank participants with whole-genome sequence.**
P-value was obtained from a linear regression of each RBC parameter with a genetic variant, adjusted for age, sex, current smoking status, and body mass index.

BreakDancer v1.3.6 to screen for common and large deletions in the α-globin gene cluster region in Taiwanese participants. When a deletion was present, the section of DNA that was absent in the participant's genome was identified through a comparison with the reference genome. When pairs from a section of DNA spanning the deletion are aligned to the genome, the inferred insert size will be larger than expected. BreakDancer uses these pairs of reads to detect deletions. Thus, the sequencing depth affects the resolution of deletion positions. The average depth of the WGS in TWB participants was ~30X. Thus, we did not anticipate to obtain numerous pairs of reads, which means that the detected location of the deletion may be approximate. In this study, we selected only

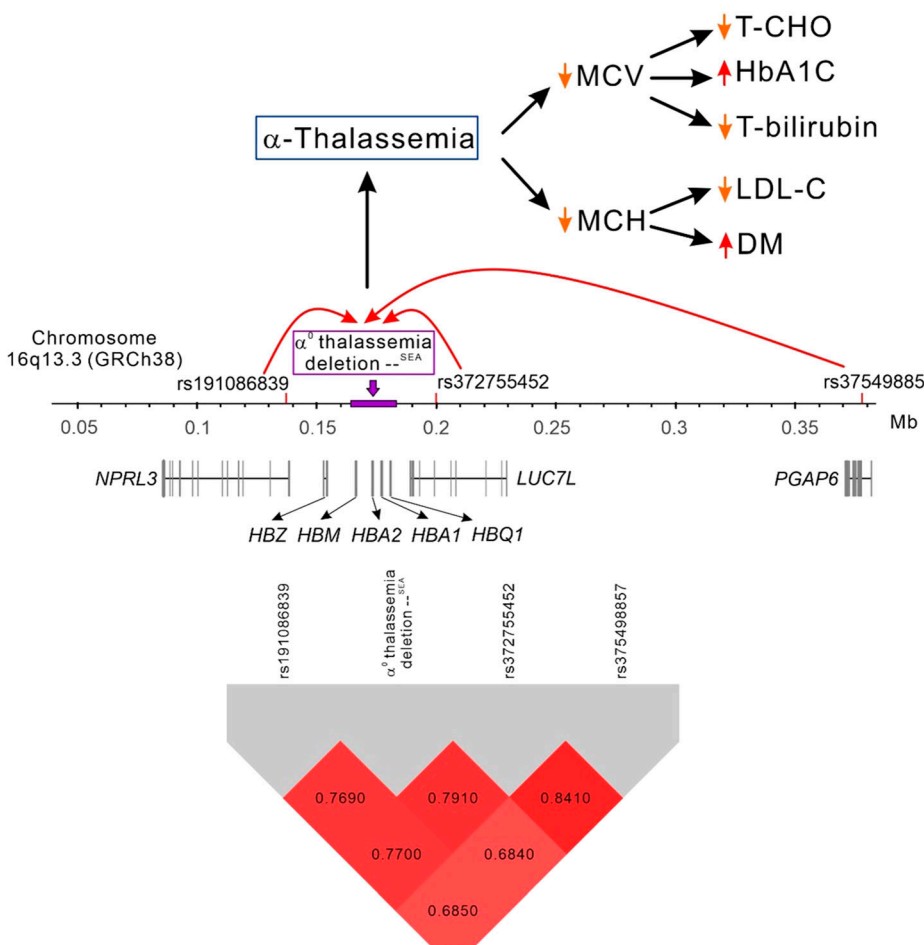

**Figure 3. Genomic structure and linkage disequilibrium map of chromosome 16p13.3 variants and their association with $\alpha^0$ thalassemia $-^{SEA}$ deletion mutation, various phenotypes, and diabetes mellitus.**

deletions involving the *HBA1* and *HBA2* gene regions for analysis. A total of 128 participants were enrolled in BreakDancer analysis, including 20 with normocytic erythrocytes and 108 with microcytic hypochromic erythrocytes. 62 participants with microcytic hypochromic erythrocytes and one participant with normocytic hypochromic erythrocytes had a large deletion involving the $\alpha$-globin gene region: 58 individuals had an ~20-kb deletion, 4 had an ~30-kb deletion, and 1 had an ~33-kb deletion (Fig 5C–F). Because the $\alpha^0$ thalassemia $-^{SEA}$ 20-kb deletion in the $\alpha$-globin gene is the most common cause of $\alpha$-thalassemia in the Taiwanese population (Chen et al, 2002; Chao et al, 2014; Lee et al, 2015; Wang et al, 2017), we conducted tests to detect the presence of this deletion in 1,474 participants who had WGS data and genomic DNA available for analysis (Fig 5G). By performing PCR, we detected the $\alpha^0$ thalassemia $-^{SEA}$ 20-kb deletion in the $\alpha$-globin gene in 60 participants, including 58 participants screened using BreakDancer (57 participants had an MCV of <80 fl and an MCH of <25 pg/RBC and one had an MCV of >80 fl and an MCH of <25 pg/RBC) and two participants who lacked BreakDancer data (one participant had an MCV of >80 fl and an MCH of <25 pg/RBC and another participant had an MCV of <80 fl and an MCH of >25 pg/RBC; Fig 5L). PCR findings confirmed that the 30-kb deletion detected by BreakDancer was because of

the $\alpha^0$ thalassemia $-^{FIL}$ deletion and the 33-kb deletion was because of the $\alpha^0$ thalassemia $-^{THAI}$ deletion (Fig 5H). The PCR results were confirmed through direct DNA sequencing (Fig 5I–K), with the exception of the 30-kb deletion mutation, in which multiple single-nucleotide repeated sequences were detected in PCR products that caused difficulty in performing DNA sequencing.

## Association between three significant SNVs on chromosome 16p13.3 and the $\alpha^0$ thalassemia $-^{SEA}$ deletion mutation detected through PCR

The $\alpha^0$ thalassemia $-^{SEA}$ deletion is the most common cause of $\alpha$-thalassemia in Taiwan. We analyzed the association among the three significant SNVs and the $\alpha^0$ thalassemia $-^{SEA}$ deletion mutation detected through PCR in 1,474 TWB participants with WGS data (Fig 5M–O). For *NPRL3* rs191086839 risk allele carrier, the sensitivity, specificity, positive predictive value (PPV), and negative predictive value (NPV) were 85.00%, 99.93%, 98.08%, and 99.37%, respectively. For *LUC7L* rs372755452 risk allele carrier, the sensitivity, specificity, PPV, and NPV were 90.00%, 99.79%, 94.74%, and 99.58%, respectively. For *PGAP6* rs375498857 risk allele carrier, the

**Table 2. Association between genetic variants from chromosome 16p13.3 and hematological parameters.**

| Variants/RBC traits | Genotypes (number) | | | Beta | SE | P-value[a] |
|---|---|---|---|---|---|---|
| *NPRL3* rs191086839 | TT (1,441) | TC (52) | CC (0) | | | |
| RBC count, $10^6$/μl | 4.74 (4.43–5.09) | 5.92 (5.59–6.28) | – | 1.1133 | 0.0529 | $2.41 \times 10^{-86}$ |
| Hemoglobin, g/dl | 14.10 (13.10–15.20) | 12.75 (11.93–13.70) | – | −1.3864 | 0.1615 | $2.30 \times 10^{-17}$ |
| Hematocrit, % | 43.80 (40.60–46.70) | 41.45 (38.98–44.68) | – | −2.5319 | 0.4949 | $3.52 \times 10^{-7}$ |
| MCV, fl | 92.13 (88.29–96.05) | 69.82 (67.49–72.62) | – | −21.6882 | 0.9372 | $1.96 \times 10^{-101}$ |
| MCH, pg/RBC | 29.90 (28.81–30.89) | 21.58 (20.01–22.32) | – | −7.9492 | 0.3067 | $1.66 \times 10^{-122}$ |
| MCHC, g/dl | 32.43 (31.14–33.42) | 31.29 (29.82–32.00) | – | −1.3718 | 0.2222 | $8.57 \times 10^{-10}$ |
| MC_HC_E, % | 4.02 | 96.15 | – | 6.5196 | 0.7430 | $1.72 \times 10^{-18}$ |
| MC_HC_A, % | 2.91 | 34.62 | – | 2.9820 | 0.3447 | $5.08 \times 10^{-18}$ |
| Anemia, % | 2.91 | 34.62 | – | 2.9618 | 0.3438 | $7.00 \times 10^{-18}$ |
| *LUC7L* rs372755452 | GG (1,436) | G- (57) | – (0) | | | |
| RBC count, $10^6$/μl | 4.74 (4.43–5.08) | 5.95 (5.61–6.29) | – | 1.1002 | 0.0501 | $8.68 \times 10^{-93}$ |
| Hemoglobin, g/dl | 14.10 (13.10–15.20) | 13.30 (12.10–13.85) | – | −1.2093 | 0.1552 | $1.25 \times 10^{-14}$ |
| Hematocrit, % | 43.70 (40.50–46.68) | 42.00 (39.70–45.60) | – | −2.0200 | 0.4749 | $2.20 \times 10^{-5}$ |
| MCV, fl | 92.14 (88.34–96.05) | 70.59 (68.00–73.22) | – | −20.4436 | 0.9018 | $5.83 \times 10^{-98}$ |
| MCH, pg/RBC | 29.91 (28.82–30.89) | 21.84 (21.10–22.50) | – | −7.5248 | 0.2950 | $2.40 \times 10^{-119}$ |
| MCHC, g/dl | 32.43 (31.14–33.42) | 31.32 (29.94–32.00) | – | −1.3014 | 0.2127 | $1.20 \times 10^{-9}$ |
| MC_HC_E, % | 3.83 | 92.98 | – | 6.0457 | 0.5575 | $2.12 \times 10^{-27}$ |
| MC_HC_A, % | 3.13 | 26.32 | – | 2.5452 | 0.3501 | $3.62 \times 10^{-13}$ |
| Anemia, % | 3.13 | 26.32 | – | 2.5452 | 0.3501 | $3.62 \times 10^{-13}$ |
| *PGAP6* rs375498857 | CC (1,437) | CA (56) | AA (0) | | | |
| RBC count, $10^6$/μl | 4.74 (4.43–5.08) | 5.95 (5.49–6.28) | – | 1.0419 | 0.0515 | $1.39 \times 10^{-80}$ |
| Hemoglobin, g/dl | 14.10 (13.10–15.20) | 13.15 (12.13–13.90) | – | −1.1910 | 0.1567 | $5.27 \times 10^{-14}$ |
| Hematocrit, % | 43.70 (40.50–46.60) | 42.25 (39.65–45.65) | – | −1.8741 | 0.4795 | $9.70 \times 10^{-5}$ |
| MCV, fl | 92.13 (88.29–96.05) | 70.67 (68.19–73.71) | – | −19.3109 | 0.9288 | $1.77 \times 10^{-84}$ |
| MCH, pg/RBC | 29.90 (28.81–30.89) | 22.01 (20.19–22.67) | – | −7.2171 | 0.3037 | $4.79 \times 10^{-106}$ |
| MCHC, g/dl | 33.43 (31.14–33.42) | 31.29 (29.89–31.98) | – | −1.3908 | 0.2142 | $1.15 \times 10^{-10}$ |
| MC_HC_E, % | 4.04 | 89.29 | – | 5.4882 | 0.4691 | $1.28 \times 10^{-31}$ |
| MC_HC_A, % | 3.20 | 25.00 | – | 2.4681 | 0.3570 | $4.73 \times 10^{-12}$ |
| Anemia, % | 3.20 | 25.00 | – | 2.4681 | 0.3570 | $4.73 \times 10^{-12}$ |

Abbreviations: RBC, red blood cell; MC_HC_E, microcytic hypochromic trait; MC_HC_A, microcytic hypochromic anemia. Other abbreviations as in Table 1.
[a]P-value using an additive model and adjusted for age, sex, BMI, and current smoking.

sensitivity, specificity, PPV, and NPV were 85.00%, 99.65%, 91.07%, and 99.37%, respectively (Table S3).

### Regional association analysis of cardiometabolic traits with chromosome 16p13.3

Among the three significant variants located on chromosome 16p13.3, only *PGAP6* rs375498857 was used in genotype–phenotype analysis performed by employing GWAS imputation data from 115,926 TWB participants for all study phenotypes (Table S4). In brief, the imputed *PGAP6* rs375498857 genotypes completely matched with genotypes derived from the WGS data of 1,493 TWB participants. Furthermore, the imputation data from the

GWAS CHB-1 Array (TWBv1.0) and the GWAS CHB-2 Array (TWBv2.0) were consistent, with 1,206 participants undergoing analysis with both the arrays. The other two variants did not meet the aforementioned criteria for imputation and thus were not included in further analysis performed using the imputation data. In addition to RBC traits, genotype–phenotype analysis revealed a genome-wide significant association of the rs375498857 genotype with total high-density lipoprotein (HDL)– and low-density lipoprotein (LDL)–cholesterol levels and HbA1c and total bilirubin levels ($P$ = $4.57 \times 10^{-36}$, $1.92 \times 10^{-12}$, $5.91 \times 10^{-24}$, $5.13 \times 10^{-40}$, and $2.25 \times 10^{-14}$, respectively; Table S4). Furthermore, the rs375498857 genotype was significantly associated with the risk of DM ($P$ = $2.00 \times 10^{-6}$). By performing a regional association study, we determined that rs375498857 is the only variant associated with the

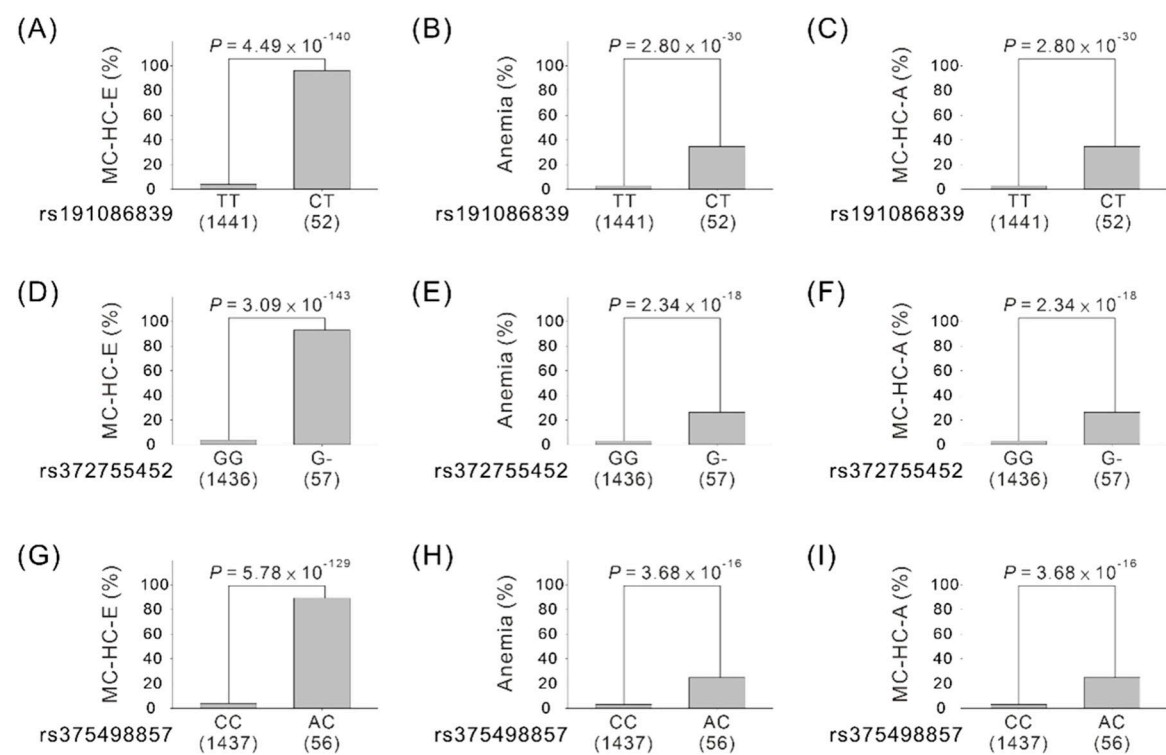

**Figure 4. Association between chromosome 16p13.3 variants and α-thalassemia indices.**
**(A, B, C, D, E, F, G, H, I)** Association between genetic variants ((A, B, C), *NPRL3* rs191086839; (D, E, F), *LUC7L* rs372755452; (G, H, I), *PGAP6* rs375498857) from chromosome 16p13.3 and microcytic hypochromic erythrocyte traits (MC-HC-E), anemia, and microcytic anemia (MC-HC-A) in 1,493 Taiwan Biobank participants with whole-genome sequencing data. *P*-values were examined using a chi-squared test.

**Table 3. Stepwise linear regression analysis for mean corpuscular volume (MCV) and mean corpuscular hemoglobin (MCH), including genotypes.**

| | MCV (fl) (1,493) | | | | MCH (pg/RBC) (1,493) | | | |
|---|---|---|---|---|---|---|---|---|
| | **Beta** | **SE** | **r²** | ***P*-value\*** | **Beta** | **SE** | **r²** | ***P*-value\*** |
| Age, years | 0.0725 | 0.0152 | 0.0112 | $1.91 \times 10^{-6}$ | 0.0349 | 0.0049 | 0.0207 | $2.49 \times 10^{-12}$ |
| Sex, male versus female | −1.0254 | 0.3618 | 0.0033 | 0.0047 | −0.8322 | 0.1180 | 0.0279 | $2.71 \times 10^{-12}$ |
| Body mass index, kg/m² | −0.1674 | 0.0474 | 0.0037 | 0.0004 | −0.0397 | 0.0155 | 0.0024 | 0.0104 |
| Current smoking, % | 2.3101 | 0.5845 | 0.0095 | 0.0001 | 0.9534 | 0.1906 | 0.0102 | $6.39 \times 10^{-7}$ |
| *NPRL3* rs191086839, TT versusTC | −12.8002 | 1.9361 | 0.2572 | $5.30 \times 10^{-11}$ | −4.5653 | 0.6315 | 0.2917 | $7.77 \times 10^{-13}$ |
| *LUC7L* rs372755452, GG versus G- | −9.6954 | 1.8529 | 0.0121 | $1.91 \times 10^{-7}$ | −3.6913 | 0.6044 | 0.0156 | $1.29 \times 10^{-9}$ |
| *PGAP6* rs375498857, CC versus CA | — | — | — | — | — | — | — | — |

aforementioned phenotypes and disease (Figs S1, S2, S3, S4, S5, S6, and S7).

## Correlation between RBC parameters and cardiometabolic traits

Various metabolic traits were significantly associated with MCV and MCH, and most of the study traits had a *P*-value threshold of $10^{-4}$. Consistent associations were observed between MCV and MCH and nearly all studied traits, with MCV generally having a stronger effect. Elevated MCV and MCH were associated with increased uric acid levels and lipid profiles, including total, HDL and LDL cholesterol and triglyceride levels. Elevated MCV reduced the risk of metabolic

syndrome and most of the metabolic syndrome-related components, such as systolic blood pressure, hypertension, fasting plasma glucose level, HbA1c level, DM status, eGFR, albuminuria, and microalbuminuria. All liver function-related test results showed the same direction with changes in MCV and MCH (Table 4).

## MR study with two-stage least square (2SLS) for the association between α-thalassemia-related erythrocyte indices and cardiometabolic traits and DM using rs375498857 variant as an IV

For MR analyses, we selected rs375498857-related cardiometabolic traits (with a *P*-value threshold of $10^{-4}$). In the 2SLS IV analysis for

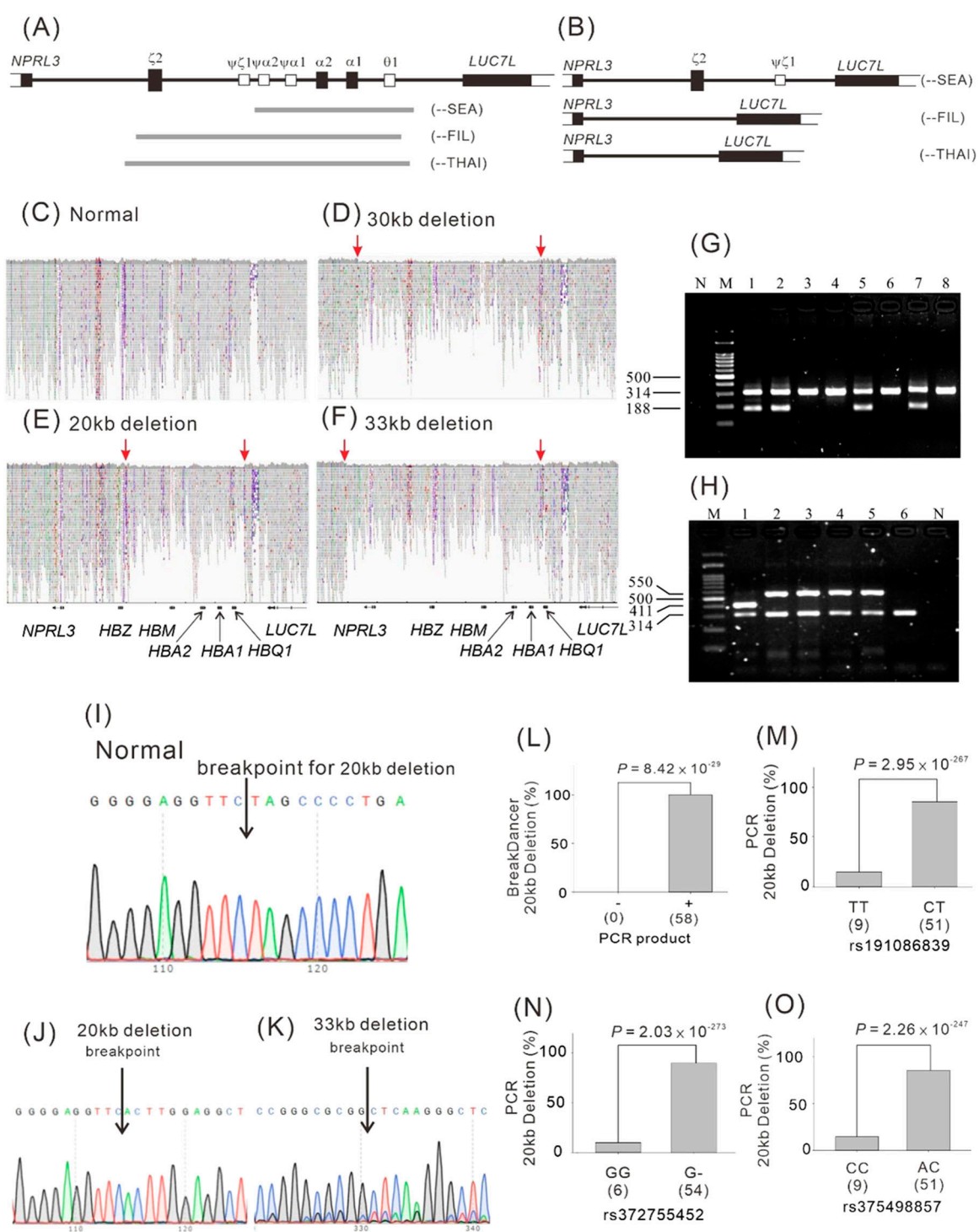

**Figure 5.  BreakDancer and PCR analysis with direct DNA sequencing for $\alpha^0$ thalassemia deletion mutations.**
**(A, B, C, D, E, F, G, H, I, J, K, L, M, N, O)** Diagrams of α-thalassemia mutations for gene alignment (A, B), genotyping by BreakDancer v1.3.6 (C, D, E, F), PCR (G, H), direct DNA sequencing (I, J, K), and association results of chromosome 16p13.3 variants (M, N, O) and genotyping performed using BreakDancer v1.3.6 (L) with the $\alpha^0$ thalassemia deletion $-^{SEA}$ detected through PCR. **(A)** Genes are represented as black boxes and pseudogenes as white boxes. The α-thalassemia mutations are represented as grey lines. **(B)** Diagrams illustrate the structure of the α-globin gene cluster on chromosome 16 with α-thalassemia mutations. **(G, H)** Agarose gels show representative results of PCR assays. **(G)** Sizes of amplified fragments are expressed in base pairs (bp). Lane N, negative control; lane M, 100-bp DNA ladder H3 RTU (GeneDireX, Inc.); (G) Lanes 1, 2, 5, and 7 indicate α-thalassemia $-^{SEA}$ heterozygotes because of the presence of the deletion-specific 188-bp band and a 314-bp band obtained from the control DNA sequence. Lanes 3, 4, 6, and 8 indicate participants without deletion mutations that provided only the 314-bp band. **(H)** Lanes 1 indicates α-thalassemia $-^{THAI}$ heterozygotes because of the presence of the deletion specific 411-bp band and a 314-bp band. Lanes 2–5 indicate α-thalassemia $-^{FIL}$ heterozygotes because of the presence of the deletion specific 550-bp band and a 314-bp band. Lanes 6 indicates participants without deletion mutation.

**Table 4.** Correlation between mean corpuscular hemoglobin (MCH) and mean corpuscular volume (MCV) with study phenotypes in Taiwan Biobank participants.

| Clinical and laboratory parameters | MCH | | | MCV | | |
|---|---|---|---|---|---|---|
| | Beta | SE | P-value[a] | Beta | SE | P-value[a] |
| Anthropology | | | | | | |
| Age, years | 0.0422 | 0.0008 | $<10^{-307}$ | 0.1227 | 0.0021 | $<10^{-307}$ |
| Sex, male/female | −0.9975 | 0.0200 | $<10^{-307}$ | −0.9989 | 0.0538 | $6.62 \times 10^{-77}$ |
| Waist circumference, cm | −0.0003 | 0.0016 | 0.8707 | 0.0160 | 0.0044 | $2.91 \times 10^{-4}$ |
| Waist–hip ratio | −0.2790 | 0.1584 | 0.0781 | −0.7899 | 0.4257 | 0.0635 |
| Body mass index kg/m$^2$ | −0.0386 | 0.0023 | $1.29 \times 10^{-65}$ | −0.1361 | 0.0061 | $1.59 \times 10^{-111}$ |
| Blood pressure | | | | | | |
| Systolic BP[a], mmHg | −0.0054 | 0.0006 | $1.02 \times 10^{-17}$ | −0.0387 | 0.0017 | $1.67 \times 10^{-116}$ |
| Diastolic BP[a], mmHg | 0.0003 | 0.0010 | 0.7440 | −0.0316 | 0.0026 | $2.09 \times 10^{-34}$ |
| Mean BP[a], mmHg | −0.0033 | 0.0009 | $1.59 \times 10^{-4}$ | −0.0424 | 0.0023 | $6.94 \times 10^{-73}$ |
| Lipid profiles | | | | | | |
| Total cholesterol#, mg/dl | 4.0048 | 0.1144 | $9.64 \times 10^{-267}$ | 7.9003 | 0.3077 | $5.69 \times 10^{-145}$ |
| HDL–cholesterol#, mg/dl | 2.7512 | 0.0961 | $1.18 \times 10^{-179}$ | 6.1504 | 0.2579 | $2.31 \times 10^{-125}$ |
| LDL–cholesterol#, mg/dl | 1.7281 | 0.0768 | $6.97 \times 10^{-112}$ | 3.7220 | 0.2061 | $8.17 \times 10^{-73}$ |
| Triglyceride#, mg/dl | 0.1958 | 0.0413 | $2.09 \times 10^{-6}$ | 0.5315 | 0.1107 | $1.56 \times 10^{-6}$ |
| Glucose metabolism | | | | | | |
| Fasting plasma glucose**, mg/dl | 0.0011 | 0.0006 | 0.0569 | −0.0092 | 0.0016 | $6.91 \times 10^{-9}$ |
| HbA1C**, % | −0.3567 | 0.0148 | $1.88 \times 10^{-128}$ | −1.3014 | 0.0396 | $6.57 \times 10^{-236}$ |
| Uric acid | | | | | | |
| Uric acid***, mg/dl | 0.1129 | 0.0078 | $5.22 \times 10^{-47}$ | 0.3026 | 0.0210 | $6.88 \times 10^{-47}$ |
| Renal function | | | | | | |
| Creatinine, mg/dl | 0.0917 | 0.0302 | 0.0024 | 0.4879 | 0.0811 | $1.78 \times 10^{-9}$ |
| eGFR, ml/min/1.73 m$^2$ | −0.0049 | 0.0004 | $3.69 \times 10^{-36}$ | −0.0152 | 0.0010 | $2.01 \times 10^{-48}$ |
| Albuminuria, mg/liter | −0.2366 | 0.0183 | $2.51 \times 10^{-38}$ | −0.4420 | 0.0491 | $2.32 \times 10^{-19}$ |
| Liver function related | | | | | | |
| AST, U/liter | 0.0146 | 0.0007 | $1.01 \times 10^{-99}$ | 0.0254 | 0.0018 | $4.90 \times 10^{-43}$ |
| ALT, U/liter | 0.0076 | 0.0004 | $1.93 \times 10^{-72}$ | 0.0134 | 0.0011 | $4.90 \times 10^{-32}$ |
| γGT, U/liter | 0.0045 | 0.0003 | $1.94 \times 10^{-64}$ | 0.0107 | 0.0007 | $1.65 \times 10^{-51}$ |
| Serum albumin, g/dl | 0.4430 | 0.0374 | $2.07 \times 10^{-32}$ | 1.1601 | 0.1004 | $7.34 \times 10^{-31}$ |
| Total bilirubin, mg/dl | 0.9540 | 0.0306 | $7.68 \times 10^{-213}$ | 1.3804 | 0.0824 | $6.80 \times 10^{-63}$ |
| Hematological parameters | | | | | | |
| Leukocyte count, 10$^3$/μl | 0.0003 | 0.0003 | 0.2698 | 0.0006 | 0.0008 | 0.4213 |
| Hematocrit, % | 0.3055 | 0.0023 | $<10^{-307}$ | 1.0471 | 0.0058 | $<10^{-307}$ |
| Platelet count, 10$^3$/μl | −0.0089 | 0.0001 | $<10^{-307}$ | −0.0193 | 0.0004 | $<10^{-307}$ |
| RBC count, 10$^6$/μl | −3.6156 | 0.0154 | $<10^{-307}$ | −10.0076 | 0.0407 | $<10^{-307}$ |
| Hemoglobin, g/dl | 1.3237 | 0.0055 | $<10^{-307}$ | 2.6878 | 0.0162 | $<10^{-307}$ |
| MCH, pg/RBC | – | – | – | 2.3700 | 0.0037 | $<10^{-307}$ |
| MCHC, g/dl | 1.0118 | 0.0046 | $<10^{-307}$ | 0.4283 | 0.0147 | $2.07 \times 10^{-186}$ |
| MCV, fl | 0.3281 | 0.0005 | $<10^{-307}$ | – | – | – |
| Atherosclerotic risk factors | | | | | | |
| Diabetes mellitus, % | −0.0454 | 0.0037 | $3.63 \times 10^{-35}$ | −0.0233 | 0.0014 | $1.54 \times 10^{-66}$ |

**Table 4. Continued**

| Clinical and laboratory parameters | MCH | | | MCV | | |
|---|---|---|---|---|---|---|
| | Beta | SE | *P*-value[a] | Beta | SE | *P*-value[a] |
| Hypertension, % | −0.0050 | 0.0029 | 0.0827 | −0.0091 | 0.0010 | $4.05 \times 10^{-18}$ |
| Current smoking, % | 0.0472 | 0.0033 | $2.52 \times 10^{-46}$ | 0.0133 | 0.0012 | $9.05 \times 10^{-30}$ |
| Gout, % | −0.0045 | 0.0061 | 0.4618 | −0.0012 | 0.0022 | 0.5812 |
| Microalbuminuria, % | −0.0347 | 0.0032 | $1.97 \times 10^{-27}$ | −0.0133 | 0.0012 | $3.11 \times 10^{-28}$ |
| Metabolic syndrome, % | −0.0040 | 0.0028 | 0.1574 | −0.0061 | 0.0010 | $4.15 \times 10^{-9}$ |

Abbreviations and participant recruitment as in Table 1 and Fig 1.
[a]Adjustment for age, sex, current smoking status, and BMI.

the direction and causality of α-thalassemia-related erythro-cyte indices and cardiometabolic traits, the association of *PGAP6* rs375498857 genotypes with cardiometabolic traits remained significant even after adjustment for multiple parameters associated with MCV or MCH (Table 5): the association between the rs375498857 genotype and LDL cholesterol levels and DM subsided after adjustment for MCH. Moreover, the association between the rs375498857 genotype and total cholesterol and total bilirubin levels and HbA1c subsided after adjustment for MCV (all *P* > 0.05). The association between the rs375498857 genotype and HDL cholesterol levels did not totally abolish after either adjustment for MCH or MCV ($P = 2.22 \times 10^{-10}$ and $P = 0.0008$, respectively). These results suggest that the association between the rs375498857 genotype and cardiometabolic traits and DM, with the exception of HDL cholesterol levels, is dependent on MCH or MCV. Moreover, consistent findings were observed in the same 2SLS analysis, using either *NPRL3* rs191086839 or *LUC7L* rs372755452 as the instrumental variable (Table S5).

# Discussion

This study explored the association of chromosome 16p13.3 variants with erythrocyte indices in the Taiwanese population. Our findings revealed that three significant SNVs at this chromosome location around the *HBA1* gene were closely associated with erythrocyte parameters, namely RBC counts, Hb, MCV, and MCH levels. TWB participants who had the minor allele of these three variants in their erythrocytes were both microcytic and hypochromic. Using Break-Dancer v1.3.6 for screening, followed by PCR amplification and direct sequencing, we confirmed that the α-thalassemia deletion mutation −SEA, the most common α-thalassemia mutation in the Taiwanese population, exhibited strong LD with these SNVs, with their NPVs ranging from 0.994 to 0.996 and PPVs from 0.912 to 0.981. Thus, these SNVs can be considered as crucial surrogate genetic markers for the α-thalassemia deletion mutation −SEA. We performed an MR study by using *PGAP6* rs375498857 as IV in participants with whole-genomic genotyping data. We observed causal relationships between MCV/MCH and rs375498857-related cardiometabolic traits and DM (Fig 3). This study is the first to demonstrate that α-thalassemia caused by the −SEA deletion mutation can be analyzed using SNVs as the surrogate marker, obviating the need to directly genotype the deletion mutation. This approach can be applied to a larger population using genotyping

analyses, such as array data, for the mass screening of α-thalassemia in adults or newborns, making it a highly powerful and effective tool for epidemiological research on α-thalassemia in Taiwan.

## Linking three significant SNVs located on chromosome 16p13.3 with a deletion mutation of α-thalassemia

Previous studies have reported that chromosome 16p13.3 as a gene locus is associated with erythrocyte traits (Kichaev et al, 2019; Vuckovic et al, 2020; Sinnott-Armstrong et al, 2021). By including TWB participants in their study, Lee et al (2022) revealed that the *PGAP6* rs375498857 genotype is associated with several RBC traits. In our study, we observed that *NPRL3* rs191086839, *LUC7L* rs372755452, and *PGAP6* rs375498857 are associated with multiple erythrocyte indices, exhibit strong LD, and are specific to the Asian population. The minor allele frequencies of all the three variants were <0.0001 in European populations but were between 0.0188 and 0.0218 in Asian populations in the 1,000 Genome Project (Table S1). Our findings indicate that the three significant SNVs may be linked to the common deletion mutation of α-thalassemia in the Taiwanese population. First, α-thalassemia is characterized by a deficit in α-globin chain synthesis and most commonly caused by the deletion in the *HBA1* gene, which is encoded by α-globin and is localized on chromosome 16p13.3 where the three SNVs are located. Second, thalassemia is prevalent in some regions of Asia and around the Mediterranean region but not in most European countries; these three SNVs are specific to the Asian population. Third, most TWB participants having the minor alleles of these three variants were both microcytic and hypochromic, with more than 90% of the participants having the microcytic hypochromic trait. Finally, more than 100 genetic forms of α-thalassemia have been identified in which $\alpha^0$-thalassemia is usually the most common clinically relevant form (Giardine et al, 2014). In Taiwan, the −SEA type of the $\alpha^0$-thalassemia mutation is the most common deletion mutation, accounting for 69–91% cases; this finding indicates that the prevalence of this mutation is ~4.0–4.5% in Taiwanese individuals (Chen et al, 2002; Lee et al, 2015; Wang et al, 2017). All the three SNPs are specific to the Asian population and highly linked to the microcytic hypochromic trait, with a minor allele frequency of ~1.77–1.94%. This translates to a heterozygous genotype frequency of ~3.5–3.8% (Table 2), which is close to the predicted prevalence of $\alpha^0$-thalassemia mutation −SEA. The associations were further confirmed using both BreakDancer screening and PCR with direct

**Table 5. Summary of coefficients used for Mendelian randomization analysis: mean corpuscular volume (MCV) and mean corpuscular hemoglobin (MCH) for cardiometabolic traits.**

| $T_A$ | $T_B$ | $G_A$ | $T_A\text{-}T_B$ Beta | SE | $P^a$ | $G_A\text{-}T_A$ Beta | SE | $P^a$ | $G_A\text{-}T_B$ Beta | SE | $P^a$ | $IV_A\text{-}T_B$ Beta | SE | $P$ | $IV_A\text{-}T_B\text{-adj}T_A$ Beta | SE | $P^b$ |
|---|---|---|---|---|---|---|---|---|---|---|---|---|---|---|---|---|---|
| MCH | Total cholesterol# (mg/dl) | PGAP6 rs375498857 | 0.0028 | 0.0001 | $9.07 \times 10^{-275}$ | −7.1828 | 0.0442 | $<10^{-307}$ | −0.0164 | 0.0013 | $4.57 \times 10^{-36}$ | 0.0023 | 0.0002 | $4.57 \times 10^{-36a}$ $(4.19 \times 10^{-37c})$ | −0.0007 | 0.0002 | $8.84 \times 10^{-4}$ |
| MCH | HDL−cholesterol# (mg/dl) | PGAP6 rs375498857 | 0.0027 | 0.0001 | $6.33 \times 10^{-183}$ | −7.1828 | 0.0442 | $<10^{-307}$ | −0.011 | 0.0016 | $1.92 \times 10^{-12}$ | 0.0015 | 0.0002 | $1.92 \times 10^{-12a}$ $(1.74 \times 10^{-8c})$ | −0.0015 | 0.0002 | $2.22 \times 10^{-10}$ |
| MCH | LDL−cholesterol# (mg/dl) | PGAP6 rs375498857 | 0.0027 | 0.0001 | $1.44 \times 10^{-117}$ | −7.1828 | 0.0442 | $<10^{-307}$ | −0.0198 | 0.002 | $5.91 \times 10^{-24}$ | 0.0028 | 0.0003 | $5.91 \times 10^{-24a}$ $(8.30 \times 10^{-25c})$ | 0.0000 | 0.0003 | 0.9206 |
| MCH | HbA1c** (%) | PGAP6 rs375498857 | −0.015 | 0.0006 | $4.88 \times 10^{-136}$ | −7.1791 | 0.0434 | $<10^{-307}$ | 0.1306 | 0.0099 | $5.13 \times 10^{-40}$ | −0.0182 | 0.0014 | $5.13 \times 10^{-40a}$ $(5.21 \times 10^{-44d})$ | −0.0043 | 0.0015 | 0.0055 |
| MCH | Total bilirubin (mg/dl) | PGAP6 rs375498857 | 0.0087 | 0.0003 | $7.47 \times 10^{-219}$ | −7.1672 | 0.0422 | $<10^{-307}$ | −0.0344 | 0.0045 | $2.25 \times 10^{-14}$ | 0.0048 | 0.0006 | $2.25 \times 10^{-14a}$ $(5.07 \times 10^{-13e})$ | −0.0049 | 0.0007 | $2.36 \times 10^{-12}$ |
| MCH | DM | PGAP6 rs375498857 | −0.0461 | 0.0036 | $1.72 \times 10^{-37}$ | −7.1672 | 0.0422 | $<10^{-307}$ | 0.2629 | 0.0558 | $2.00 \times 10^{-6}$ | −0.0367 | 0.0078 | $2.00 \times 10^{-6a}$ $(4.75 \times 10^{-4d})$ | 0.0122 | 0.0089 | 0.1682 |
| MCV | Total cholesterol# (mg/dl) | PGAP6 rs375498857 | 0.0008 | 0.00003 | $2.34 \times 10^{-150}$ | −18.3275 | 0.1198 | $<10^{-307}$ | −0.0164 | 0.0013 | $4.57 \times 10^{-36}$ | 0.0009 | 0.0001 | $4.57 \times 10^{-36a}$ $(4.19 \times 10^{-37c})$ | 0.0001 | 0.0001 | 0.0621 |
| MCV | HDL−cholesterol# (mg/dl) | PGAP6 rs375498857 | 0.0009 | 0.00004 | $4.88 \times 10^{-127}$ | −18.3275 | 0.1198 | $<10^{-307}$ | −0.011 | 0.0016 | $1.92 \times 10^{-12}$ | 0.0006 | 0.0001 | $1.92 \times 10^{-12a}$ $(1.74 \times 10^{-8c})$ | −0.0003 | 0.0001 | 0.0008 |
| MCV | LDL−cholesterol# (mg/dl) | PGAP6 rs375498857 | 0.0008 | 0.00004 | $6.74 \times 10^{-77}$ | −18.3275 | 0.1198 | $<10^{-307}$ | −0.0198 | 0.002 | $5.91 \times 10^{-24}$ | 0.0011 | 0.0001 | $5.91 \times 10^{-24a}$ $(8.30 \times 10^{-25c})$ | 0.0003 | 0.0001 | 0.0069 |
| MCV | HbA1c** (%) | PGAP6 rs375498857 | −0.0075 | 0.0002 | $3.90 \times 10^{-245}$ | −18.3175 | 0.1178 | $<10^{-307}$ | 0.1306 | 0.0099 | $5.13 \times 10^{-40}$ | −0.0071 | 0.0005 | $5.13 \times 10^{-40a}$ $(5.21 \times 10^{-44d})$ | 0.0004 | 0.0006 | 0.4670 |
| MCV | Total bilirubin (mg/dl) | PGAP6 rs375498857 | 0.0018 | 0.0001 | $1.50 \times 10^{-66}$ | −18.2699 | 0.1148 | $<10^{-307}$ | −0.0344 | 0.0045 | $2.25 \times 10^{-14}$ | 0.0019 | 0.0002 | $2.25 \times 10^{-14a}$ $(5.07 \times 10^{-13e})$ | 0.0002 | 0.0003 | 0.5457 |
| MCV | DM | PGAP6 rs375498857 | −0.0233 | 0.0013 | $3.28 \times 10^{-69}$ | −18.2699 | 0.1148 | $<10^{-307}$ | 0.2629 | 0.0558 | $2.00 \times 10^{-6}$ | −0.0144 | 0.0031 | $2.43 \times 10^{-6a}$ $(4.75 \times 10^{-4d})$ | 0.0114 | 0.0034 | $8.00 \times 10^{-4}$ |

$T_A$ and $T_B$: phenotypes A (MCV and MCH level) and B (Total cholesterol, HDL–C, LDL–C, HbA1c, total bilirubin, and DM); $G_A$, genotypes determining $T_A$; $IV_A$, instrumental variables for $G_A$.
aAdjustment for age, sex, current smoking status, and BMI.
bAdjustment for MCV or MCH level.
cAdjustment for age, sex, current smoking status, BMI, and other possible confounders, such as systolic blood pressure (BP), hemoglobin A1C (HbA1c), uric acid (UA), estimated glomerular filtration rate (eGFR), aspartate aminotransferase (AST), and total bilirubin.
dAdjustment for age, sex, current smoking status, BMI, and other possible confounders, such as systolic BP, total cholesterol, UA, eGFR, AST, and total bilirubin.
eAdjustment for age, sex, current smoking status, BMI, and other possible confounders, such as systolic BP, total cholesterol, HbA1c, UA, eGFR, and AST.

sequencing, which revealed considerably high NPV (0.9937–0.9958) and PPV (0.9107–0.9808) values for the three SNVs. With the increasing health and economic burden of thalassemia in recent years because of population growth and epidemiologic transition, identifying surrogate genetic markers can be helpful for future mass screening and developing preventive medicine strategies for thalassemia.

### The *NPRL3* rs191086839 variant is the strongest and independent genetic surrogate marker for the $\alpha^0$-thalassemia deletion mutation $-^{SEA}$ in Taiwanese individuals

*NPRL3* is a highly conserved gene located upstream of the *HBA* gene cluster. The α-globin locus of all mammalian species analyzed lies within a region of 135–155 kb of conserved synteny, with α-like genes arranged along the chromosome in the order 5′-ξ-α-α-3′ (Fig 3). The *HBA* cluster is located between *NPRL3* and *LUC7L* genes in almost all mammals except mouse, in which the *HBA* cluster no longer has *LUC7L* downstream of the globin genes (Vernimmen, 2014; Philipsen & Hardison, 2018). The *NPRL3* gene contains the enhancers of the *HBA* gene cluster. The erythroid-specific multispecies conserved sequences (MCSs) identified by DNase-hypersensitive sites are numbered from MCS-R1 to MCS-R4 (Hughes et al, 2005). Three of these elements (MCS-R1, MCS-R2, and MCS-R3) lie within the body, and MCS-R4 lies upstream of the promoter of the *NPRL3* gene. MCS-R2 has multiple roles, and these roles may be applicable to any other enhancer: the recruitment of polymerase II and key transcription factors at the promoter, formation of a looped structure involving several remote regulatory elements, and removal of repressive complexes, such as PcG. By performing functional analysis, Miyata et al (2020) demonstrated that an erythroid-specific enhancer is located in the intron 7 of vertebrate *NPRL3*, which indicates the presence of a remote enhancer on nprl3 in multiglobin gene expression. In multivariate analysis, *NPRL3* rs191086839 was observed to have the strongest effect on various erythrocyte indices, including MCV and MCH, and the microcytic hypochromic trait and microcytic anemia (Tables 3 and S2). Therefore, considering the robust evidence of co-inheritance, as reflected in the linkage disequilibrium analysis and the context of surrogate markers traditionally being associated with sensitivity and specificity, we propose *NPRL3* rs191086839 as a strong and independent candidate genetic surrogate marker for the $\alpha^0$-thalassemia deletion mutation $-^{SEA}$ in Taiwanese individuals.

### Association between α-thalassemia-related erythrocyte indices and metabolic traits

Our findings revealed that an elevated MCV was associated with lower risks of metabolic syndrome and some of its components and complications, such as DM, hypertension, microalbuminuria, and lower HDL cholesterol levels. These results are compatible with those reported previously indicating that an elevated MCV was associated with lower risks of metabolic syndrome and visceral obesity (Tanaka et al, 2020, 2021). Metabolic syndrome and its related components have been reported to be associated with various adverse cardiovascular and cancer outcomes (Lakka et al, 2002; Emery et al, 2022). In addition, a high MCV was determined to be associated with elevated total and LDL cholesterol, uric acid, and

triglyceride levels in our study. Previous studies have reported that macrocytosis was associated with the severity or poor prognosis of various cardio-renal diseases (Solak et al, 2013; Ueda et al, 2013; Hsieh et al, 2017; Wang et al, 2020). Similar trends of associations were noted between MCH and most of the metabolic traits analyzed. These results suggest the diverse and bidirectional effects of MCV and MCH, which result in both favorable and unfavorable cardiometabolic outcomes. Additional studies may be necessary to elucidate the effect of the interaction between erythrocyte indices and various cardiometabolic risk factors on the outcome of cardiovascular diseases.

### Causal relationships among α-thalassemia-related RBC traits, metabolic traits, and DM

Previous GWASs have revealed associations between chromosome 16p13.3 variants and HbA1c and LDL cholesterol levels (Sinnott-Armstrong et al, 2021; Hsu et al, 2022; Lee et al, 2022). Thus, we examine the association between *PGAP6* rs375498857 and various metabolic and biochemical traits in 115,926 TWB participants with whole-genome genotyping array data. We observed that the rs375498857 genotype was closely related to many cardiometabolic traits, such as lipid profiles (total cholesterol, LDL cholesterol, and HDL cholesterol levels), HbA1c levels, and total bilirubin levels, and DM. By performing an MR study, we determined that the correlation between this SNV and cardiometabolic traits (low total and LDL cholesterol levels) is mainly mediated by erythrocyte traits, such as MCV and MCH. This study is the first to indicate the importance of α-thalassemia-related erythrocyte traits in causing cardiometabolic traits and increasing DM risk (OR = 1.01–1.04); however, exact mechanisms underlying these associations are not fully understood. Total and LDL cholesterol levels in $\beta^0$-thalassemia carriers (thalassemia minors) were lower than those in age- and sex-matched controls (Maioli et al, 1989). The LDL-lowering effect of $\beta^0$-thalassemia was proposed to be because of (1) reactive mild erythroid hyperplasia increasing LDL removal by the bone marrow and (2) the chronic activation of the monocyte–macrophage system resulting in the increased secretion of some cytokines that affect hepatic secretion and receptor-mediated removal of apolipoprotein B-containing lipoproteins (Deiana et al, 2000). The latter mechanism resembles mild chronic inflammation and has been linked to DM development through an increase in oxidative stress and insulin resistance. Moreover, our study confirms the findings of earlier reports indicating that having the α-thalassemia trait significantly increases the likelihood of developing gestational diabetes among Chinese women (Lao & Ho, 2001). Moreover, individuals with α-thalassemia in Iran had a higher risk of impaired glucose tolerance (either prediabetes or diabetes) than did the general population (Bahar et al, 2015). Therefore, health-care providers should screen individuals with α-thalassemia for these conditions and provide interventions to reduce their risk, including lifestyle modifications and medication. The causal relationship sheds light on underlying mechanisms that may be studied in the future, such as how variations in RBC production and function affect inflammation, oxidative stress, and endothelial dysfunction, all of which are linked to the emergence of cardiometabolic traits and DM. Future large transethnic prospective studies should be conducted to

determine the roles of these SNPs in thalassemia screening and epidemiology and their association with cardiometabolic disorders.

### Limitation

Because of the existence of ethnic heterogeneity, the results of this study should be interpreted with caution when applying them to different races. This includes investigating whether the relationship between SNVs and deletion mutations is the same among different races and whether other deletion mutations can be identified using matching SNVs.

In conclusion, our results revealed that three Asian-specific chromosome 16p13.3 variants, namely *NPRL3* rs191086839, *LUC7L* rs372755452, and *PGAP6* rs375498857, exhibit strong LD and can be used as surrogate genetic markers for α-thalassemia or the $\alpha^0$ thalassemia $-^{SEA}$ deletion, with high specificity and sensitivity. *PGAP6* rs375498857 in GWAS arrays revealed a significant association with multiple metabolic traits. MR indicated that many α-thalassemia-related metabolic traits, such as total and LDL cholesterol levels, HbA1c, and DM, are causally related to erythrocyte traits, such as MCV and MCH. These results may be beneficial in the future preventive medicine for population screening and understanding the underlying genetic causes of thalassemia-related cardiometabolic traits and DM.

## Materials and Methods

### TWB participants

The TWB recruited adults from centers across Taiwan between 2008 and 2020. The initial cohort for the regional association study of erythrocyte indices consisted of 1,494 TWB participants who underwent WGS; one individual from this cohort was excluded because of incomplete WGS data. Subsequently, to expand the study cohort, we included 129,542 participants who underwent genotyping with Axiom Genome-Wide CHB Arrays for a regional association study. From this cohort, 13,616 participants were excluded for the following reasons: the data of 7,216 participants were excluded as quality control (QC) for the GWAS, 4,647 participants fasted for <6 h, and 1,753 participants had failed genotyping for the rs375498857 polymorphism. During QC for the GWAS, all excluded participants were related as second-degree relatives or closer, with an identity-by-descent value of >0.187. Participants with a history of hypertension, hyperlipidemia, DM, and gout were excluded when respective parameters were analyzed. Hypertension, DM, obesity, current smoking, microalbuminuria, and metabolic syndrome were defined as reported previously (Wu et al, 2022). This study was approved by the Research Ethics Committee of Taipei Tzu Chi Hospital, Buddhist Tzu Chi Medical Foundation (approval numbers: 05-X04-007 and 10-XD-056) and the Ethics and Governance Council of the TWB (approval numbers: TWBR10507-02 and TWBR10611-03). Each participant was asked to sign an approved informed consent form.

### DNA extraction and genotyping

Genomic DNA from blood samples was isolated using the PerkinElmer Chemagic 360 instrument (PerkinElmer). SNV genotyping

was performed either through WGS or by using custom TWB chips. SNV genotyping was performed using the Axiom Genome-Wide Array Plate System (Affymetrix).

### Hematological indices

The following hematological parameters were analyzed: RBC count and white blood cell count, platelet count, and hematocrit (Hct) and hemoglobin (Hb) concentrations. Briefly, blood cell indices were calculated using a hematology analyzer, which provided complete count data. Other parameters were calculated from the same indices, as follows: MCV, which was calculated as Hct divided by the RBC count, and MCH, which was calculated as the Hb concentration divided by the RBC count. The microcytic hypochromic trait was considered when MCV was <80 fl and MCH was <25 pg. Microcytic anemia was defined as the microcytic hypochromic trait with Hct <40% for men and <36% for women or Hb < 13 g/dl for men and <12 g/dl for women.

### Regional association studies performed using TWB WGS data for gene regions surrounding the HBA1 gene

To determine the association of the significant SNVs located on chromosome 16p13.3 with hematological indices, we performed a regional association analysis of the WGS data of TWB participants (Fig 1). WGS data from a subgroup of TWB participants were evaluated through an ultrafast whole-genome secondary analysis on Illumina sequencing platforms with Illumina HiSeq 2500 or Illumina Hiseq 4000 sequencers (Raczy et al, 2013). The resulting reads were aligned to the hg19 reference genome with iSAAC 01.13.10.21. iSAAC Variant Caller 2.0.17 was used to perform SNP and insertion–deletion variant discovery and genotyping (Raczy et al, 2013). A total of 1,493 participants with 18,198 SNVs located at positions between 0.06 and 0.68 Mb on chromosome 16p13.3 were recruited. An in-house protocol written in shell script was used to combine 1,493 vcf files. A union table of all detected variants was used for regional association analysis. The association between SNPs and clinical and laboratory parameters was analyzed using the GWAS method.

### Detection of deletion mutation by using deletion mutation detection software BreakDancer

The bam files of WGS data were provided by TWB (Juang et al, 2021). The reference genome version used was hg19. Samtools v1.3.1 (Li et al, 2009) was used to select the region of interest from original WGS bam files. BreakDancer v1.3.6 (Chen et al, 2009) was used to detect long deletion mutations in the region of interest.

### Polymerase chain reaction amplification and direct sequencing

To detect candidate α-thalassemia deletion mutations, the WT and $\alpha^0$ thalassemia deletion $-^{SEA}$ sequences were genotyped through PCR by using the primer pairs (forward) 5'-GCGATCTGGGCTCTGTGTTCT-3', (reverse) 5'-GTTCCCTGAGCCCCGACACG-3' (Chang et al, 1991), and (reverse) 5'-GCCTTGAACTCCTGGACTTAA-3', respectively (Sanguansermsri et al, 1999). The $\alpha^0$ thalassemia deletion $-^{THAI}$ sequence was genotyped through PCR amplification by using

the primer pairs (forward) 5'-CACGAGTAAAACATCAAGTACACTCCAGCC-3' and (reverse) 5'-TGGATCTGCACCTCTGGGTAGGTTCTGTACC-3'. The $\alpha^0$ thalassemia deletion $-^{FIL}$ sequence was genotyped through PCR amplification by using the primer pairs (forward) 5'-AAGAGAA-TAAACCACCCAATTTTTAAATGGGCA-3' and (reverse) 5'-GAGATAA-TAACCTTTATCTGCCACATGTAGCAA-3' (Liu et al, 2000).

### Clinical phenotypes and laboratory examinations

To determine cardiometabolic traits, we examined the following clinical phenotypes: age; BMI; waist circumference; waist–hip ratio; and systolic, mean, and diastolic blood pressure. We also collected the following biochemistry data: lipid profiles, namely total, HDL, and LDL cholesterol and triglyceride levels; glucose metabolism parameters, namely fasting plasma glucose and HbA1c levels; and liver and renal functional test–related parameters, namely serum creatinine, uric acid, aspartate aminotransferase, alanine amino-transferase, γ-glutamyl transferase, albumin, and total bilirubin levels. BMI and the estimated glomerular filtration rate were calculated as reported previously (Hsu et al, 2021). Because of the absence of data on the urine creatinine level, only the spot urine albumin level was used to evaluate albuminuria. The hematological parameters analyzed also included white blood cell and platelet counts.

### Regional association analysis of cardiometabolic traits

We performed regional association studies including the Axiom Genome-Wide CHB array data of participants, as reported previously (Hsu et al, 2022; Yeh et al, 2022). A total of 115,926 participants were included in the analysis.

### Statistical analysis

Categorical data are presented as frequencies and numbers. Continuous variables are expressed as medians and interquartile ranges. Lipid parameters and urine albumin levels were logarithmically transformed when analyzed using regression. A general linear regression was performed in the association study between the phenotypes and genotypes, with adjustment for potential confounders, such as age, sex, BMI, and current smoking status. Logistic regression was performed to investigate the effect of variants on microcytic traits, anemia, and other risk factors. Stepwise multivariable regression was performed to determine the independent correlates of hematological parameters. Statistical analyses were performed using IBM SPSS Statistics for Windows (Version 22; SPSS). We used PLINK (version 1.07; Shaun Purcell, https://zzz.bwh.harvard.edu/plink/, accessed on 3 January, 2022) for regional association analysis. LocusZoom (http://csg.sph.umich.edu/locuszoom/, accessed on 4 January, 2022) was employed for drawing regional plots. A P-value of < $5 \times 10^{-8}$ was defined as indicating genome-wide significance. For P-values between $5 \times 10^{-8}$ and $1 \times 10^{-4}$, a subthreshold that suggested GWAS significant association of SNVs was considered. The subthreshold locus has been demonstrated to be effective for predicting studied phenotypes confirmed using luciferase reporter assays (Wang et al, 2016). Linkage disequilibrium (LD) between SNVs was analyzed using LDmatrix software (https://ldlink.nih.gov/?tab=ldmatrix, accessed on 4 January, 2022).

### MR approaches

We performed an instrumental variable (IV) regression analysis by using 2SLS methods to examine whether rs375498857, which is a α-thalassemia-related MCV and MCH-determining genetic variant, is associated with other cardiometabolic traits and DM development through their associations with MCV or MCH. The 2SLS method is widely used for evaluating continuous and binary exposures and outcomes (Bowden et al, 2016; Burgess et al, 2017; Hsu et al, 2022). In standard MR, the first stage involved regressing MCV- and MCH-determining SNV to generate predicted MCV and MCH values. The second stage involved regressing study parameters on rs375498857 to predict the risk of cardiometabolic traits. The strength of the instruments was assessed using the F statistic that was calculated using the equation $F = R^2(n - 2)/(1 - R^2)$, where $R^2$ is the proportion of the variability in genetically determined MCV or MCH accounted for by the SNV, and n is the sample size (Palmer et al, 2012). An F statistic of >10 indicates a relatively low risk of weak instrument bias in MR analyses (Palmer et al, 2012). Statistical power for 2SLS MR was calculated using the noncentrality parameter-based approach described previously (Brion et al, 2013).

## Data Availability

The data presented in this study are available on request from the corresponding author.

### Ethics statement

This study was approved by the Research Ethics Committee of Taipei Tzu Chi Hospital, Buddhist Tzu Chi Medical Foundation (approval numbers: 05-X04-007 and 10-XD-056) and the Ethics and Governance Council of the TWB (approval numbers: TWBR10507-02 and TWBR10611-03). Each participant was asked to sign an approved informed consent form.

## Supplementary Information

## Acknowledgements

We greatly appreciate the technical support of the Core Laboratory of the Taipei Tzu Chi Hospital, Buddhist Tzu Chi Medical Foundation, expert statistical analysis assistance from Tsung-Han Hsieh, and Kimforest LTD Taiwan for bioinformatics support. This study was supported by grants from Buddhist Tzu Chi Medical Foundation (TCMF-EP 111-02), grants from the Ministry of Science and Technology (MOST 108-2314-B-303 -026 -MY3), and Taipei Tzu Chi Hospital, Buddhist Tzu Chi Medical Foundation (TCRD-TPE-112-02) to Y-L Ko.

## Author Contributions

L-A Hsu: formal analysis, validation, methodology, and writing—original draft, review, and editing.
S Wu: software, validation, methodology, and writing—review and editing.
M-S Teng: methodology.
Y-L Ko: conceptualization, resources, formal analysis, supervision, funding acquisition, validation, project administration, and writing—original draft, review, and editing.

## Conflict of Interest Statement

The authors declare that they have no conflict of interest.

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
