## [Reviewer comments · Life Science Alliance]

Life Science Alliance

Causal links of α -thalassemia indices and cardiometabolic traits and diabetes: MR study

Lung-An Hsu, Semon Wu, Ming-Sheng Teng, and Yu-Lin Ko
DOI: <https://doi.org/10.26508/lsa.202302204>

Corresponding author(s): Yu-Lin Ko, Taipei Tzu Chi Hospital

Review Timeline:

Submission Date:	2023-06-07
Editorial Decision:	2023-07-21
Revision Received:	2023-08-08
Editorial Decision:	2023-09-11
Revision Received:	2023-09-15
Accepted:	2023-09-19

Scientific Editor: Novella Guidi

Transaction Report:

July 21, 2023

Re: Life Science Alliance manuscript #LSA-2023-02204-T

Yu-Lin Ko
Taipei Tzu Chi Hospital

Dear Dr. Ko,

Thank you for submitting your manuscript entitled "Causal associations between α -thalassemia-related erythrocyte indices and cardiometabolic traits and diabetes mellitus susceptibility: A Mendelian randomization study" to Life Science Alliance. The manuscript was assessed by an expert reviewer, whose comments are appended to this letter. We invite you to submit a revised manuscript addressing the Reviewer comments.

When submitting the revision, please include a letter addressing the reviewer comments point by point.

Thank you for this interesting contribution to Life Science Alliance. We are looking forward to receiving your revised manuscript.

Sincerely,

B. MANUSCRIPT ORGANIZATION AND FORMATTING:

Reviewer #1 (Comments to the Authors (Required)):

This manuscript studied 16p13.3 region, covering a known deletion for alpha-thalassemia (monogenic disorder), for its genetic association with blood-related traits (RBC count, MCV, MCH, HB) and cardiometabolic traits in Taiwanese population. They found three lead SNPs (single nucleotide polymorphism) associated with blood-related traits and those SNPs are highly correlated with -SEA deletion, suggesting being used as a surrogate marker. They further conducted a stepwise linear regression with 3 SNPs on MCV and MCH. Additionally, they performed the association of genetic variants on 16p13.3 with cardiometabolic traits using the imputed genetic data, followed by a Mendelian randomization. I found several concerns.

I have three major comments.

1. Regarding the stepwise linear regression, given that -SEA is the causal genetic factor and 3 SNPs and -SEA are all in high LD, the stepwise linear regression to identify independent variants might be meaningless.
2. Line 275. Why only PGAP6 was used/included in GWAS imputed data analysis? Is it only available in the GWAS imputed data? Even though, two other variants do not completely agree/match with WGS data, if their imputation quality is good, we want to see the results of them. Also, I don't think you applied the same exclusion criteria on other variants (other than those 3 variants) in imputed data. Also, it looks like PGAP6 has the lowest priority from the previous sections.
3. I might have missed but it is unclear why MCV and MCH were selected to be examined more with cardiometabolic traits. It is not clear what's the goal of MR analysis in this manuscript. Is it for the causal relationship between RBC parameters and cardiometabolic traits? If so, you need to use all genetic variants associated with each trait and bidirectional MR can be performed. If you are interested in examining alpha-thalassemia triggered cascade of RBC parameters and cardiometabolic traits, then, a mediation analysis (rs375498877 -> RBC -> cardiometabolic trait; rs375498877 -> cardiometabolic trait -> RBC) looks more appropriate.

In Abstract, it will be good to mention -SEA mutation (in the first sentence), which can easily justify to perform regional association at 16p13.3.

Fig 3. What's the rationale or any biological mechanism to draw the directed diagram between alpha-thalassemia, MCV, MCH and other traits (total cholesterol, etc)?

Table 3. Why only conducted only for MCV and MCH? As the homozygous form minor-alleles did not present in this data, remove those in the label (e.g., TT vs TC vs CC -> TT vs TC).

Pg 8. Line 134 - what do you mean by specific to Asians? It looks like it means that the alternative allele is very rare in other Asian. Please clarify in the sentence and show those allele frequencies and -SEA mutation rate in this study and in other race. How did you define 3 SNPs as lead snps?

Pg 16. Line 261: is this sensitivity for -SEA deletion detection by NPRL3 rs191086839 risk allele carrier? Please make it clear.

In Table 5, G_A - T_B and IV_A - T_B have the same p-value when using the same covariates, as expected. This probably indicates that MR is not a good choice for this.

Response to reviewer 1: We greatly appreciate the recommendations of the reviewer.

1. Regarding the stepwise linear regression, given that -SEA is the causal genetic factor and 3 SNPs and -SEA are all in high LD, the stepwise linear regression to identify independent variants might be meaningless.

Response: Thank you for this important comment. Because these three SNPs are situated in distinct genes, their biological relevance to the outcomes of interest is unknown, and they are not in complete LD, the issues associated with multicollinearity are less severe. Despite redundancy, our intention is to demonstrate that the association between *PGAP6* rs375498857 and the erythrocyte traits is primarily attributed to the strong LD with *NPRL3* rs191086839 and *LUC7L* rs372755452.

2. Line 275. Why only *PGAP6* was used/included in GWAS imputed data analysis? Is it only available in the GWAS imputed data? Even though, two other variants do not completely agree/match with WGS data, if their imputation quality is good, we want to see the results of them. Also, I don't think you applied the same exclusion criteria on other variants (other than those 3 variants) in imputed data. Also, it looks like *PGAP6* has the lowest priority from the previous sections.

Response: The imputation and quality control procedures were performed following the methods described in the reference (Hsu et al. 2022). We applied the same exclusion criteria for all variants in the imputed data. The decision to use only *PGAP6* in the GWAS imputed data analysis was explained in the original version. Among the 1493 participants with WGS data, 1427 also underwent GWAS CHB-1 Array, and their imputed *PGAP6* rs375498857 genotypes matched perfectly with those derived from the WGS data. However, for the other two SNPs, there were 2 participants each whose genotypes did not match those derived from the WGS data. Similarly, among the 1206 participants who underwent analysis with both the TWB1 and TWB2 arrays, the imputed *PGAP6* rs375498857 data from the GWAS CHB-1 Array (TWBv1.0) and the GWAS CHB-2 Array (TWBv2.0) were completely consistent. However, for the other two variants, there were three participants whose imputed genotypes did not match between the two arrays. To address the reviewer's concern, we included MR analysis results using *NPRL3* rs191086839 or *LUC7L* rs372755452 as instrumental variables in supplementary Table 5. The main findings were consistent with the results obtained from *PGAP6* rs375498857.

Supplementary Table 5. Summary of coefficients used for Mendelian randomization analysis: MCV and MCH for cardiometabolic traits

(A) *NPRL3* rs191086839 as instrumental variable

T _A	T _B	G _A	T _A -T _B			G _A -T _A			G _A -T _B			IV _A -T _B			IV _A -T _B -adjT _A		
			Beta	SE	P ^a	Beta	SE	P ^a	Beta	SE	P ^a	Beta	SE	P	Beta	SE	P ^b
MCH	Total cholesterol# (mg/dL)	NPRL3 rs191086839	0.0028	0.0001	9.07×10^{-275}	-7.61191	0.04712	$< 10^{-307}$	-0.01847	0.00142	1.16×10^{-38}	0.00243	0.00019	1.16×10^{-38a} (1.62×10^{-40c})	-0.00054	0.00021	0.00907
MCH	HDL-cholesterol# (mg/dL)	NPRL3 rs191086839	0.0027	0.0001	6.33×10^{-183}	-7.61191	0.04712	$< 10^{-307}$	-0.01369	0.00170	6.96×10^{-16}	0.00180	0.00022	6.96×10^{-16a} (3.62×10^{-11c})	-0.00126	0.00025	4.12×10^{-7}
MCH	LDL-cholesterol# (mg/dL)	NPRL3 rs191086839	0.0027	0.0001	1.44×10^{-117}	-7.61191	0.04712	$< 10^{-307}$	-0.02299	0.00212	2.71×10^{-27}	0.00302	0.00028	2.71×10^{-17a} (8.52×10^{-29c})	0.00032	0.00031	0.30603
MCH	HbA1c** (%)	NPRL3 rs191086839	-0.0150	0.0006	4.88×10^{-136}	-7.62092	0.04620	$< 10^{-307}$	0.14389	0.01068	2.37×10^{-41}	-0.01888	0.00140	2.37×10^{-41a} (1.16×10^{-45d})	-0.00542	0.00157	0.00054
MCH	Total bilirubin (mg/dL)	NPRL3 rs191086839	0.0087	0.0003	7.47×10^{-219}	-7.61262	0.04499	$< 10^{-307}$	-0.03900	0.00490	1.79×10^{-15}	0.00512	0.00064	1.79×10^{-15a} (3.79×10^{-14e})	-0.00471	0.00072	5.44×10^{-11}
MCH	DM	NPRL3 rs191086839	-0.0461	0.0036	1.72×10^{-37}	-7.61262	0.04499	$< 10^{-307}$	0.25961	0.06063	1.90×10^{-5}	-0.03410	0.00796	0.00002^a (0.00236^d)	0.01607	0.00908	0.07666

T _A	T _B	G _A	T _A -T _B			G _A -T _A			G _A -T _B			IV _A -T _B			IV _A -T _B -adjT _A		
			Beta	SE	P ^a	Beta	SE	P ^a	Beta	SE	P ^a	Beta	SE	P	Beta	SE	P ^b
MCV	Total cholesterol# (mg/dL)	NPRL3 rs191086839	0.0008	0.00003	2.34×10^{-150}	-19.48948	0.12792	$< 10^{-307}$	-0.01847	0.00142	1.16×10^{-38}	0.00095	0.00007	1.16×10^{-38a} (1.62×10^{-40c})	0.00021	0.00008	0.00956
MCV	HDL-cholesterol# (mg/dL)	NPRL3 rs191086839	0.0009	0.00004	4.88×10^{-127}	-19.48948	0.12792	$< 10^{-307}$	-0.01369	0.00170	6.96×10^{-16}	0.00070	0.00009	6.96×10^{-16a} (3.62×10^{-11c})	-0.00020	0.00010	0.03388
MCV	LDL-cholesterol# (mg/dL)	NPRL3 rs191086839	0.0008	0.00004	6.74×10^{-77}	-19.48948	0.12792	$< 10^{-307}$	-0.02299	0.00212	2.71×10^{-27}	0.00118	0.00011	2.71×10^{-17a} (8.52×10^{-29c})	0.00043	0.00012	0.00034
MCV	HbA1c** (%)	NPRL3 rs191086839	-0.0075	0.0002	3.90×10^{-245}	-19.49675	0.12570	$< 10^{-307}$	0.14389	0.01068	2.37×10^{-41}	-0.00738	0.00055	2.37×10^{-41a} (1.16×10^{-45d})	0.00011	0.00060	0.85735
MCV	Total bilirubin (mg/dL)	NPRL3 rs191086839	0.0018	0.0001	1.50×10^{-66}	-19.46998	0.12258	$< 10^{-307}$	-0.03900	0.00490	1.79×10^{-15}	0.00200	0.00025	1.79×10^{-15a} (3.79×10^{-14e})	0.00034	0.00028	0.22792
MCV	DM	NPRL3 rs191086839	-0.0233	0.0013	3.28×10^{-69}	-19.46998	0.12258	$< 10^{-307}$	0.25961	0.06063	1.90×10^{-5}	-0.01333	0.00311	0.00002^a (0.00236^d)	0.01302	0.00348	0.00019

(B) *LUC7L* rs372755452 as instrumental variable

T _A	T _B	G _A	T _A -T _B			G _A -T _A			G _A -T _B			IV _A -T _B			IV _A -T _B -adjT _A		
			Beta	SE	P ^a	Beta	SE	P ^a	Beta	SE	P ^a	Beta	SE	P	Beta	SE	P ^b

MCH	Total cholesterol# (mg/dL)	LUC7L rs372755452	0.0028	0.0001	9.07×10^{-275}	-7.52015	0.04366	$< 10^{-307}$	-0.01734	0.00132	3.51×10^{-39}	0.00231	0.00018	3.51×10^{-39a} (1.34×10^{-40c})	-0.00069	0.00020	0.00048
MCH	HDL-cholesterol# (mg/dL)	LUC7L rs372755452	0.0027	0.0001	6.33×10^{-183}	-7.52015	0.04366	$< 10^{-307}$	-0.01233	0.00158	6.25×10^{-15}	0.00164	0.00021	6.25×10^{-15a} (2.33×10^{-10c})	-0.00147	0.00024	6.11E-10
MCH	LDL-cholesterol# (mg/dL)	LUC7L rs372755452	0.0027	0.0001	1.44×10^{-117}	-7.52015	0.04366	$< 10^{-307}$	-0.02156	0.00198	1.33×10^{-27}	0.00287	0.00026	1.33×10^{-27a} (7.52×10^{-29c})	0.00015	0.00030	0.62043
MCH	HbA1c** (%)	LUC7L rs372755452	-0.0150	0.0006	4.88×10^{-136}	-7.51768	0.04286	$< 10^{-307}$	0.14354	0.00996	4.95×10^{-47}	-0.01909	0.00133	4.95×10^{-47a} (5.41×10^{-52d})	-0.00554	0.00150	0.00022
MCH	Total bilirubin (mg/dL)	LUC7L rs372755452	0.0087	0.0003	7.47×10^{-219}	-7.51255	0.04171	$< 10^{-307}$	-0.03640	0.00457	1.66×10^{-15}	0.00485	0.00061	1.66×10^{-15a} (7.34×10^{-14c})	-0.00505	0.00069	1.69E-10
MCH	DM	LUC7L rs372755452	-0.0461	0.0036	1.72×10^{-37}	-7.51255	0.04171	$< 10^{-307}$	0.25824	0.05649	5.00×10^{-6}	-0.03437	0.00752	5.00×10^{-6a} (0.00083^d)	0.01567	0.00871	0.07199
T _A	T _B	G _A	T _A -T _B			G _A -T _A			G _A -T _B			IV _A -T _B			IV _A -T _B -adjT _A		
			Beta	SE	P ^a	Beta	SE	P ^a	Beta	SE	P ^a	Beta	SE	P	Beta	SE	P ^b
MCV	Total cholesterol# (mg/dL)	LUC7L rs372755452	0.0008	0.00003	2.34×10^{-150}	-19.21820	0.11866	$< 10^{-307}$	-0.01734	0.00132	3.51×10^{-39}	0.0009	0.00007	3.51×10^{-39a} (1.34×10^{-40c})	0.00016	0.00008	0.04386
MCV	HDL-cholesterol# (mg/dL)	LUC7L rs372755452	0.0009	0.00004	4.88×10^{-127}	-19.21820	0.11866	$< 10^{-307}$	-0.01233	0.00158	6.25×10^{-15}	0.00064	0.00008	6.25×10^{-15a} (2.33×10^{-10c})	-0.00028	0.00009	0.00233
MCV	LDL-cholesterol# (mg/dL)	LUC7L rs372755452	0.0008	0.00004	6.74×10^{-77}	-19.21820	0.11866	$< 10^{-307}$	-0.02156	0.00198	1.33×10^{-27}	0.00112	0.00010	1.33×10^{-27a} (7.52×10^{-29c})	0.00037	0.00012	0.00140
MCV	HbA1c** (%)	LUC7L rs372755452	-0.0075	0.0002	3.90×10^{-245}	-19.19834	0.11673	$< 10^{-307}$	0.14354	0.00996	4.95×10^{-47}	-0.00748	0.00052	4.95×10^{-47a} (5.41×10^{-52d})	0.00003	0.00058	0.95796
MCV	Total bilirubin (mg/dL)	LUC7L rs372755452	0.0018	0.0001	1.50×10^{-66}	-19.17470	0.11376	$< 10^{-307}$	-0.03640	0.00457	1.66×10^{-15}	0.0019	0.00024	1.66×10^{-15a} (7.34×10^{-14c})	0.00023	0.00027	0.39803
MCV	DM	LUC7L rs372755452	-0.0233	0.0013	3.28×10^{-69}	-19.17470	0.11376	$< 10^{-307}$	0.25824	0.05649	5.00×10^{-6}	-0.01347	0.00295	5.00×10^{-6a} (0.00083^d)	0.01298	0.00334	0.00010

T_A and T_B: phenotypes A (MCV and MCH level) and B (Total cholesterol, HDL-C, LDL-C, HbA1c, total bilirubin, and DM); G_A: genotypes determining T_A; IV_A: instrumental variables for G_A.

a: Adjustment for age, sex, current smoking status, and BMI

b: Adjustment for MCV or MCH level.

c: Adjustment for age, sex, current smoking status, BMI, and other possible confounders, such as systolic blood pressure (BP), hemoglobin A1C (HbA1c), uric acid (UA), estimated glomerular filtration rate (eGFR), aspartate aminotransferase (AST), and total bilirubin.

d: Adjustment for age, sex, current smoking status, BMI, and other possible confounders, such as systolic BP, total cholesterol, UA, eGFR, AST and total bilirubin.

e: Adjustment for age, sex, current smoking status, BMI, and other possible confounders, such as systolic BP, total cholesterol, HbA1c, UA, eGFR and AST.

3. I might have missed but it is unclear why MCV and MCH were selected to be examined more with cardiometabolic traits. It is not clear what's the goal of MR analysis in this manuscript. Is it for the causal relationship between RBC parameters and cardiometabolic traits? If so, you need to use all genetic variants associated with each trait and bidirectional MR can be performed. If you are interested in examining alpha-thalassemia triggered cascade of RBC parameters and cardiometabolic traits, then, a mediation analysis (rs375498877 -> RBC -> cardiometabolic trait; rs375498877 -> cardiometabolic trait -> RBC) looks more appropriate.

Response: Thank you for this important comment. Since we initially conducted a regional association analysis (16p13.3) to identify the genetic variants used as proxies for α -thalassemia, our study aims to determine the causal relationship between α -thalassemia-related erythrocyte indices—specifically MCV and MCH, chosen based on their most significant association with 16p13.3 and impact on erythrocyte count and Hb level—and cardiometabolic traits. We identified the genetic variant rs375498877, robustly associated with α -thalassemia-related MCV and MCH, and further assessed its association with cardiometabolic traits. Subsequently, we used rs375498877 as an instrumental variable to estimate the causal effect of α -thalassemia-related MCV and MCH on cardiometabolic traits (IV_A - T_B). In contrast, mediation analysis aims to identify intermediary processes explaining the relationship between α -thalassemia-related MCV and MCH and cardiometabolic traits within a specific dataset. Mediation analysis assesses the effect of MCV and MCH on cardiometabolic traits while controlling for the mediator (e.g., rs375498877). This is not the case in our study. To avoid misunderstanding, we have replaced the term "RBC parameters" with " α -thalassemia-related erythrocyte indices" in the revised manuscript (highlighted in red).

In Abstract, it will be good to mention -SEA mutation (in the first sentence), which can easily justify to perform regional association at 16p13.3.

Response: We have mentioned it in the revised Abstract section (in the first sentence). Alpha-thalassemia, a microcytic anemia caused by reduced α -globin synthesis, is common in Southeast Asian and Chinese populations due to the (--^{SEA}) mutation.

Fig 3. What's the rationale or any biological mechanism to draw the directed diagram between alpha-thalassemia, MCV, MCH and other traits (total cholesterol, etc)?

Response: We have added the rationale in the revised Result section: MCV and MCH were thus selected for in-depth examination with cardiometabolic traits based on their most significant association with 16p13.3 and their impact on erythrocyte count and Hb level.

Table 3. Why only conducted only for MCV and MCH? As the homozygous form minor-alleles did not present in this data, remove those in the label (e.g., TT vs TC vs CC -> TT vs TC).

Response: MCV and MCH were selected for in-depth examination with cardiometabolic traits based on their most significant association with 16p13.3 and their impact on erythrocyte count and Hb level. Additionally, we have taken away the labels for the homozygous minor-alleles genotype.

Pg 8. Line 134 - what do you mean by specific to Asians? It looks like it means that the alternative allele is very rare in other Asian. Please clarify in the sentence and show those allele frequencies and -SEA mutation rate in this study and in other race. How did you define 3 SNPs as lead snps?

Response: As already shown in supplemental Table 1, the minor allele frequencies of these 3 SNVs among European, American, and African populations are all <0.0001. Lead SNVs are defined and selected as the SNVs with the lowest p-value for the targeted region exhibiting genome-wide significant associations. We have clarified these two issues in the revised version as follows: Our findings revealed that three lead SNVs (with the smallest, i.e., most significant, p-values in this region), namely *NPRL3* rs191086839, *LUC7L* rs372755452, and *PGAP6* rs375498857, were specific

to Asians (with minor allele frequencies ranging from 0.0177 to 0.0218 among TWB and east Asian populations vs. all <0.0001 among other ethnic populations).

Pg 16. Line 261: is this sensitivity for -SEA deletion detection by *NPRL3* rs191086839 risk allele carrier? Please make it clear.

Response: To avoid confusion, we rewrote it as follow: For *NPRL3* rs191086839 risk allele carrier, the sensitivity, specificity, positive predictive value (PPV), and negative predictive value (NPV) were 85.00%, 99.93%, 98.08%, and 99.37%, respectively. For *LUC7L* rs372755452 risk allele carrier, the sensitivity, specificity, PPV, and NPV were 90.00%, 99.79%, 94.74%, and 99.58%, respectively. For *PGAP6* rs375498857 risk allele carrier, the sensitivity, specificity, PPV, and NPV were 85.00%, 99.65%, 91.07%, and 99.37%, respectively.

In Table 5, G_A - T_B and IV_A - T_B have the same p-value when using the same covariates, as expected. This probably indicates that MR is not a good choice for this.

Response: Thank you for this important comment. The association between the genetic variant of the exposure variable (G_A) and the outcome variable (T_B) reflects the observational or epidemiological association between the exposure and the outcome. In MR, this association is used as evidence to inform causal inference about the effect of the exposure on the outcome. On the other hand, the association between the instrumental variable (IV_A) and the outcome variable (T_B) reflects the total causal effect of the exposure on the outcome. If the association between the genetic variant of the exposure variable (G_A) and the outcome variable (T_B) has the same p-value as the association between the instrumental variable (IV_A) and the outcome (T_B), it implies that the genetic variant is a valid instrument for the exposure variable in MR analysis. Specifically, it suggests that the genetic variant is strongly associated with the exposure of interest and, in turn, influences the outcome through the exposure.

September 11, 2023

RE: Life Science Alliance Manuscript #LSA-2023-02204-TR

Dr. Yu-Lin Ko
Taipei Tzu Chi Hospital
The Division of Cardiology, department of Internal Medicine, Cardiovascular Medical Center
289 Jianguo Road, Xindian City, New Taipei City 231
New Taipei
Taiwan

Dear Dr. Ko,

Thank you for submitting your revised manuscript entitled "Causal links of α -thalassemia indices and cardiometabolic traits and diabetes: MR study". We would be happy to publish your paper in Life Science Alliance pending final revisions necessary to meet our formatting guidelines.

- please address the Reviewer's remaining comments
- please exclude figures from the manuscript text and upload them separately
- please upload all figure files as individual ones, including the supplementary figure files; all figure legends should only appear in the main manuscript file
- please add the Twitter handle of your host institute/organization as well as your own or/and one of the authors in our system
- please add your main, supplementary figure, and table legends to the main manuscript text after the references section

A. FINAL FILES:

B. MANUSCRIPT ORGANIZATION AND FORMATTING:

Sincerely,

Reviewer #1 (Comments to the Authors (Required)):

The manuscript has been improved, but clarify of manuscript can be still improved as the manuscript covers multiple topics using different types of data (surrogate marker, lead snps with RBC traits and cardiometabolic traits, MR). Especially, the abstract and introduction of study objective (at the end of introduction) can be polished as it was not easy to follow the overall flow of the study and which data was utilized for each step.

Some minor editing comments:

Line 118. Rather than calling them lead SNPs, it is more accurate to call them significant SNPs (p

Line 123. "genome-wide significant association" : is this "genome-wide" or on "16p13.3"?

Figure 2 legend. (line 134) "P-value adjusted for xxx" -> p-value was obtained from a linear regression of each RBC parameter with a genetic variant, adjusted for age,

In the figure, r2 coloring does not match with r2 in figure 3. Also, in line 128-129, is this r2 from the data of this study or from other reference database?

Line 211. This sentence reads like that there are 1474 participants with -SEA deletion.

Line 407-408. Isn't the surrogate marker -SEA be based on sensitivity, rather than the association with MCV/MCH?

Response to reviewer 1:

1. The manuscript has been improved, but clarify of manuscript can be still improved as the manuscript covers multiple topics using different types of data (surrogate marker, lead snps with RBC traits and cardiometabolic traits, MR). Especially, the abstract and introduction of study objective (at the end of introduction) can be polished as it was not easy to follow the overall flow of the study and which data was utilized for each step.

Response: We would like to express our sincere gratitude for the incredibly detailed review of our manuscript. Your thoughtful and meticulous examination of our work has provided invaluable insights and suggestions for improvement, which we truly appreciate.

The abstract has been rewritten to enhance the clarity of our study's extensive content and to make it easier for readers to follow the overall flow of the study and understand which data were utilized at each step, as follows: Our study aimed to investigate if genetic variants around 16p13.3's *HBA1* locus, associated with erythrocyte indices and HbA1c levels, predict α -thalassemia-related erythrocyte indices, cardiometabolic traits, and diabetes risk in Taiwanese individuals. We analyzed Taiwan Biobank data, including whole-genome sequencing (WGS) from 1493 participants and genotyping arrays from 129,542 individuals. First, we performed regional association analysis using WGS data to identify genetic variants significantly associated with erythrocyte indices, confirming their linkage disequilibrium with the α^0 thalassemia --^{SEA} deletion mutation, a common cause of α -thalassemia in Southeast Asian populations. Deletion mutation sequencing further validated these variants' association with α -thalassemia. Subsequently, we analyzed genotyping array data, revealing associations between specific genetic variants and cardiometabolic traits, including lipid profiles, HbA1c levels, bilirubin levels, and diabetes risk. Utilizing Mendelian randomization, we established causal relationships between α -thalassemia-related erythrocyte indices and cardiometabolic traits, elucidating their role in diabetes susceptibility. Our findings highlight genetic variants around the α -globin genes as surrogate markers for common α -thalassemia mutations in Taiwan, emphasizing the causal links between α -thalassemia-related erythrocyte indices, cardiometabolic traits, and heightened diabetes risk.

The description of the introduction of study objective has also been adjusted as follow: Our study was primarily designed to uncover and elucidate the causal relationships between α -thalassemia-related erythrocyte indices, cardiometabolic

traits, and DM. We initiated by conducting a regional association analysis within the 16p13.3 region and performed deletion mutation sequencing on WGS data to pinpoint genetic variants serving as proxies for α -thalassemia. Subsequently, our investigation expanded to encompass genotyping array data from an extensive cohort of 129,542 participants. This enabled us to comprehensively explore associations between specific genetic variants and a spectrum of cardiometabolic traits, along with assessing the risk of DM. In the final phase, we harnessed MR to estimate the causal effect of α -thalassemia-related erythrocyte indices on various cardiometabolic traits and DM, thereby enhancing our understanding of these intricate interactions.

2. Line 118. Rather than calling them lead SNPs, it is more accurate to call them significant SNPs ($p < xx$) for RBC parameters as they are in LD or were not tested for independence yet.

Response: To address the reviewer's concern, we have deleted the "lead".

3. Line 123. "genome-wide significant association" : is this "genome-wide" or on "16p13.3"?

Response: It is on 16p13.3. Our initial description was not clear and caused confusion. Our intention was to convey that the association reached the genome-wide significance threshold. We have since removed it.

4. Figure 2 legend. (line 134) "P-value adjusted for xxx" -> p-value was obtained from a linear regression of each RBC parameter with a genetic variant, adjusted for age,

Response: We have added this description.

5. In the figure, r^2 coloring does not match with r^2 in figure 3. Also, in line 128-129, is this r^2 from the data of this study or from other reference database?

Response: Thank you for your attention to these detailed findings. Figure 3, which displays the LD map, reveals a strong linkage disequilibrium among these three variants in our study population. This figure was generated using Prism's Heat Map function, with the r^2 values calculated by PLINK. In Figure 2, the regional plots were created by inputting our regional association analysis results into the LocusZoom tool (<http://locuszoom.org/>). The LD data for this tool was obtained from its database, and

it only allows the selection of a general Asian population without specifying Chinese ethnicity. This is why you observed discrepancies in the coloring.

6. Line 211. This sentence reads like that there are 1474 participants with -SEA deletion.

Response: Thanks, we have clarified this sentence as follow: we conducted tests to detect the presence of this deletion in 1474 participants who had WGS data and genomic DNA available for analysis.

7. Line 407-408. Isn't the surrogate marker -SEA be based on sensitivity, rather than the association with MCV/MCH?

Response: Thank you for this important comment. As per the reviewer's observation, the utilization of the SNP as a surrogate marker for --^{SEA} is based on its co-inheritance with the disease gene (--^{SEA}). This co-inheritance is supported by strong evidence from both the LD analysis and sensitivity/specificity tests. We acknowledge that our initial description may have been unclear, leading to confusion. Our intention was to convey that among the three SNPs strongly linked with the --^{SEA} mutation and potentially usable as its surrogate marker, NPRL3 rs191086839 emerged as the strongest and independent candidate genetic predictor of MCV and MCH following a multivariate stepwise regression analysis. Furthermore, our analysis finds support in previous functional annotation studies of this chromosomal region, which encompasses the α -globin locus, as mentioned in the Discussion. To address the reviewer's concern, we have adjusted our description as follow: "Therefore, considering the robust evidence of co-inheritance, as reflected in the linkage disequilibrium analysis and the context of surrogate markers traditionally being associated with sensitivity and specificity, we propose NPRL3 rs191086839 as a strong and independent candidate genetic surrogate marker for the α^0 -thalassemia deletion mutation --^{SEA} in Taiwanese individuals."

September 19, 2023

RE: Life Science Alliance Manuscript #LSA-2023-02204-TRR

Dr. Yu-Lin Ko
Taipei Tzu Chi Hospital
The Division of Cardiology, department of Internal Medicine, Cardiovascular Medical Center
289 Jianguo Road, Xindian City
New Taipei, Taiwan

Dear Dr. Ko,

Thank you for submitting your Research Article entitled "Causal links of α -thalassemia indices and cardiometabolic traits and diabetes: MR study". It is a pleasure to let you know that your manuscript is now accepted for publication in Life Science Alliance. Congratulations on this interesting work.

DISTRIBUTION OF MATERIALS:

Again, congratulations on a very nice paper. I hope you found the review process to be constructive and are pleased with how the manuscript was handled editorially. We look forward to future exciting submissions from your lab.

Sincerely,
